# Molecular insights into the capsular polysaccharide transporter Wza-Wzc complex

Biao Yuan [1] ✉, Christian Sieben [2], Prateek Raj[1], Tina Rietschel [3], Rory Hennell James [4,5,6], Anja Gatzemeier[1], Lothar Jänsch[3], Thomas C. Marlovits [4,5,6] & Dirk W. Heinz [1] ✉

Capsular polysaccharides (CPS) are key virulence determinants, constituting the protective capsule that surrounds bacterial pathogens. Here, we present the complete cryo-EM structure of Gram-negative bacterial CPS secretion machinery formed by the *E. coli* K12 Wza-Wzc complex. The structure reveals an elongated, continuous channel spanning the entire envelope that facilitates CPS secretion. Multiple structural snapshots of the ADP-bound Wza-Wzc complex capture intermediate conformations of the double membrane assembly, highlighting its remarkable intrinsic dynamics. In-depth analysis of the isolated Wza translocon and Wzc co-polymerase, reveals mechanistic details of both complex formation and CPS transport. We further uncover the jellyroll domain of Wzc as a CPS-binding module, likely guiding CPS repeat units into a proposed Wzc-Wzy polymerization platform. Collectively, this work provides structural and functional insights into CPS synthesis and transport, advancing our understanding of bacterial capsule formation and virulence mechanisms.

The bacterial capsule acts as a thick protective layer around the cell wall, enhancing the bacterium's ability to withstand environmental stresses. It plays a crucial role in pathogenesis, being involved in virulence, immune evasion, antibiotic resistance, and biofilm formation[1]. In addition, some of the key components of the capsule, such as capsular polysaccharides (CPS), have significant potential for medical and biotechnological applications[2,3]. CPS synthesis primarily occurs through four assembly pathways involving the Wzx/Wzy-, ABC transporter-, synthase-, or extracellular transglycosylase-dependent mechanisms[4,5]. Of these, the Wzx/Wzy-dependent assembly pathway is the most commonly utilized in both Gram-negative and Gram-positive bacteria, not only for CPS but also for other high molecular weight carbohydrates, such as lipopolysaccharides (LPS), enterobacterial common antigen (ECA), and exopolysaccharides (EPS)[4,6]. *E. coli* CPS K30 and EPS colanic acid are also produced through the Wzx/Wzy-

dependent pathway[4,7]. The CPS repeat unit is synthesized in the cytoplasm by a variety of glycosyltransferases, followed by translocation into the periplasm by the flippase Wzx and transfer to the CPS polymerase Wzy for polymerization[6,8–11].

The CPS co-polymerase Wzc is an inner membrane tyrosine autokinase that regulates CPS chain length during polymerization[12,13]. Recent cryo-EM structures of *E. coli* K30 Wzc$_{K540M}$, which mimics the dephosphorylated state of the Wzc oligomer, have provided structural insights into its octameric form and domain architecture[12,13]. Specifically, the kinase domain contains the Walker-A ATP-binding motif, which works in conjunction with the tyrosine-rich tail (Y-tail) to facilitate autophosphorylation of these tyrosines[12,13]. The cognate Wzb phosphatase then promotes Wzc oligomerization by dephosphorylating its tyrosine-rich tail, which is localized at the oligomeric interfaces[14,15].

[1]Department of Molecular Structural Biology, Helmholtz Centre for Infection Research (HZI), Braunschweig, Germany. [2]Nanoscale Infection Biology Group, Department of Cell Biology, Helmholtz Centre for Infection Research (HZI), Braunschweig, Germany. [3]Cellular Proteome Research Group, Helmholtz Centre for Infection Research (HZI), Braunschweig, Germany. [4]Institute for Microbial and Molecular Sciences, University Medical Center Hamburg-Eppendorf (UKE), Hamburg, Germany. [5]Centre for Structural Systems Biology (CSSB), Hamburg, Germany. [6]Deutsches Elektronen-Synchrotron Zentrum (DESY), Hamburg, Germany. ✉e-mail: biao.yuan@helmholtz-hzi.de; dirk.heinz@helmholtz-hzi.de

The jellyroll domain, located between the inner membrane and the periplasmic helical arm domain, has an as-yet-unknown function. In contrast, the periplasmic helical arm domain plays a crucial role for CPS secretion, likely mediating the recruitment of the outer membrane Wza translocon[13]. Orthologous systems in Gram-positive bacteria, such as the CpsC-CpsD protein pair in *Streptococcus pneumoniae*, lack this translocon component, as CPS secretion in these organisms does not require an outer membrane translocon[16,17].

The coupled secretion machinery is encoded by the *wzabc* operon in *E. coli* K30 strain, which produces the K30 CPS composed of repeating units featuring a →2)-α-D-Manp-(1→3)-β-D-Galp-(1→ backbone, with a β-D-GlcA-(1→3)-α-D-Galp-(1→ disaccharide branch linked to the 3-position of the backbone mannose[7,18]. Wza can form a stable alpha-helical, "amphora"-like octamer, which is essential for CPS export[19]. The non-phosphorylated Wzc likewise adopts an octameric state, and its interaction with the Wza translocon, validated by in vivo cross-linking and negative stain EM analysis, suggests that Wza and Wzc may form a complex for CPS export[13,20,21].

Despite this, the detailed architecture of the proposed CPS secretion machinery remains unknown. To address this gap, we aimed to provide structural insights into its assembly and the mechanism of CPS secretion. We utilized the orthologous secretion system of colanic acid, which can functionally complement K30 secretion[7]. For simplicity, we refer to colanic acid as CPS hereafter.

In this work, we show by cryo-EM that Wzc and Wza assemble into a large secretion complex spanning the inner and outer bacterial membranes. Furthermore, the structure establishes a continuous translocation channel bridging the entire periplasmic space. Multiple conformational states of Wzc, captured both in the presence and absence of Wza, unveil dynamic structural changes involved in the assembly and disassembly of the CPS secretion machinery, marked by significant structural rearrangements of Wzc's periplasmic helical arms towards the Wza translocon. In addition, our findings uncover a glycan-recognition role for Wzc's periplasmic jellyroll domain, providing the molecular basis into the coordinated transfer of CPS repeat units through the Wza-Wzc complex.

## Results

### The Wza-Wzc complex forms a continuous secretion channel

The Wza-Wzc complex has been proposed to be essential for CPS secretion, although with limited in vivo evidence[13,21,22]. To address the necessity of an assembled Wza-Wzc complex for CPS secretion in vivo, we conducted fluorescence microscopy on live *E. coli* BL21 Star (DE3) cells, expressing the entire *E. coli* K12 *wzabc* operon, with Wza-mScarlet3 and Wzc-sfGFP expressed under low IPTG induction (40 μM). Wza appeared as distinct clusters localized within the membrane, consistent with previous observations for the orthologous Wza from *E. coli* K30 strain[19,21]. By contrast, Wzc exhibited two distinct distribution patterns: localized clusters and an even membrane-distribution signal, suggesting the presence of multiple organizational states. Notably, colocalization was observed between clustered forms of Wza and Wzc with no noticeable spatial bias (Fig. 1a), suggesting that these CPS transporters are uniformly distributed in the cell envelope. The D1 domain of Wza has previously been predicted to mediate interactions with Wzc[13]. Therefore, we generated a deletion mutant Wza$_{\Delta D1}$ as a negative control. Although Wza$_{\Delta D1}$ was still able to assemble into well-defined clusters, the markedly low Pearson's correlation coefficient (PCC = 0.1) confirmed a complete disruption of the Wza-Wzc interaction (Fig. 1b, c). To exclude potential artifacts from protein overexpression, we assessed colocalization at reduced IPTG concentrations (10 μM, 20 μM). Although expression was reduced at lower IPTG concentration (Supplementary Fig. 1a, b), a similar trend was observed under both conditions, demonstrating that the colocalization is not attributable to overexpression (Fig. 1c).

Wza forms a well-characterized, stable octameric complex that functions as an outer membrane CPS translocon[19]. Previous studies have shown that the Wza octamer is highly resistant to SDS treatment[23] and exhibits a pronounced tendency to form two-dimensional crystalline arrays[22]. These properties indicate that Wza can exist not only as discrete octameric units but also as higher-order assemblies. In line with these findings, our fluorescence microscopy analysis of Wza clusters reveals a broad distribution of fluorescence intensities (Fig. 1d). The predominant population exhibits relative intensities around 1, most likely representing single octameric complexes. A smaller yet distinct subset displays higher intensities (ranging from 2 to 5), suggesting higher-order assemblies. This finding aligns with the known crystallization propensity of Wza and supports the interpretation that the observed clusters represent both stable octamers and supramolecular arrangements.

To further understand the structural architecture of the CPS secretion machinery, we reconstructed the *wzabc* operon under a T7 promoter with the original ribosome binding site sequences to maintain a near-native protein expression ratio and introduced a Strep-tag at the C-terminus of Wza. Only a small amount of Wzc and Wzb was co-purified with Wza (Supplementary Fig. 1c), consistent with our live-cell imaging data showing a substantial portion of Wzc was evenly distributed in the inner membrane and did not colocalize with Wza (Fig. 1a). By introducing the K540M mutation to *E. coli* K12 Wzc, we were able to purify a stable *E. coli* K12 Wza-Wzc$_{K540M}$ complex (Supplementary Fig. 1d, e). K540 is a central residue within the conserved Walker-A motif responsible for ATP binding and hydrolysis[13,24,25]. Mutating it to methionine leads to stable Wzc octamer formation[13,24,25].

Cryo-EM analysis of the Wza-Wzc$_{K540M}$ complex, with a global spatial resolution of 3.5 Å (without symmetry applied) and 3.2 Å (applying C8 symmetry), revealed a well-defined, channel-shaped assembly spanning both the inner and outer membranes (Fig. 2a and Supplementary Fig. 2). These reconstructions demonstrated the highly symmetric nature of the complex with a length of approximately 360 Å, composed of the outer membrane Wza translocon (140 Å) and the inner membrane tyrosine auto-kinase Wzc$_{K540M}$ (220 Å). This state is designated as the Conformation 0 (Conf 0), representing the ADP-Mg$^{2+}$ bound Wza-Wzc complex. Segmented representation showed a central cavity, characteristic of the CPS secretion machinery (Fig. 2b). In the periplasmic region (Fig. 2c), the complex comprised Wza domains D1-D3 and the helical arm (HA) domain, consisting of an upper arm and a forearm, and a jellyroll (JR) domain of Wzc. The extended HA forearm interacts with the Wza D1 domain to form a continuous CPS transport path. The ADP-Mg$^{2+}$ bound tyrosine auto-kinase (Y-kinase) domain forms the cytoplasmic region of the complex, and mediates its disassembly via auto-phosorylation[15].

To investigate how these structural features of Wzc influence CPS production, we conducted site-directed mutagenesis on wild type Wzc targeting four critical regions: the Wza-Wzc (AC) interface (D99-K332, E102-Y334, E102-H338, Y155-T335), the JR domain, charged residues at the channel surface (E354, R374), and charged residues within the channel (D301, K307) (Fig. 2d, e). To assess the functional impact of the mutations on CPS production, we performed a standard colanic acid extraction and quantification assay[25] using an *E. coli* JM109 (DE3) Δ*wzabc* strain complemented with plasmids expressing the *wzabc* operon carrying the respective Wzc variants with a C-terminal Strep-tag. All tested Wzc variants exhibited expression levels similar to those observed for the wild-type protein (Supplementary Fig. 1f). As shown in Fig. 2f, quantitative analysis revealed differential levels of CPS production among the Wzc variants. Notably, the AC interface mutations (Wzc$_{K332E,Y334A,T335A,H338E}$) completely abolished CPS production, with the charged residue pairs D99-K322 and E102-H338 contributing to the electrostatic interactions, underscoring the essential role of the Wza-Wzc complex in the CPS production. In contrast, substitutions at Y334 and T335 had little to no effect, with the latter even enhancing CPS

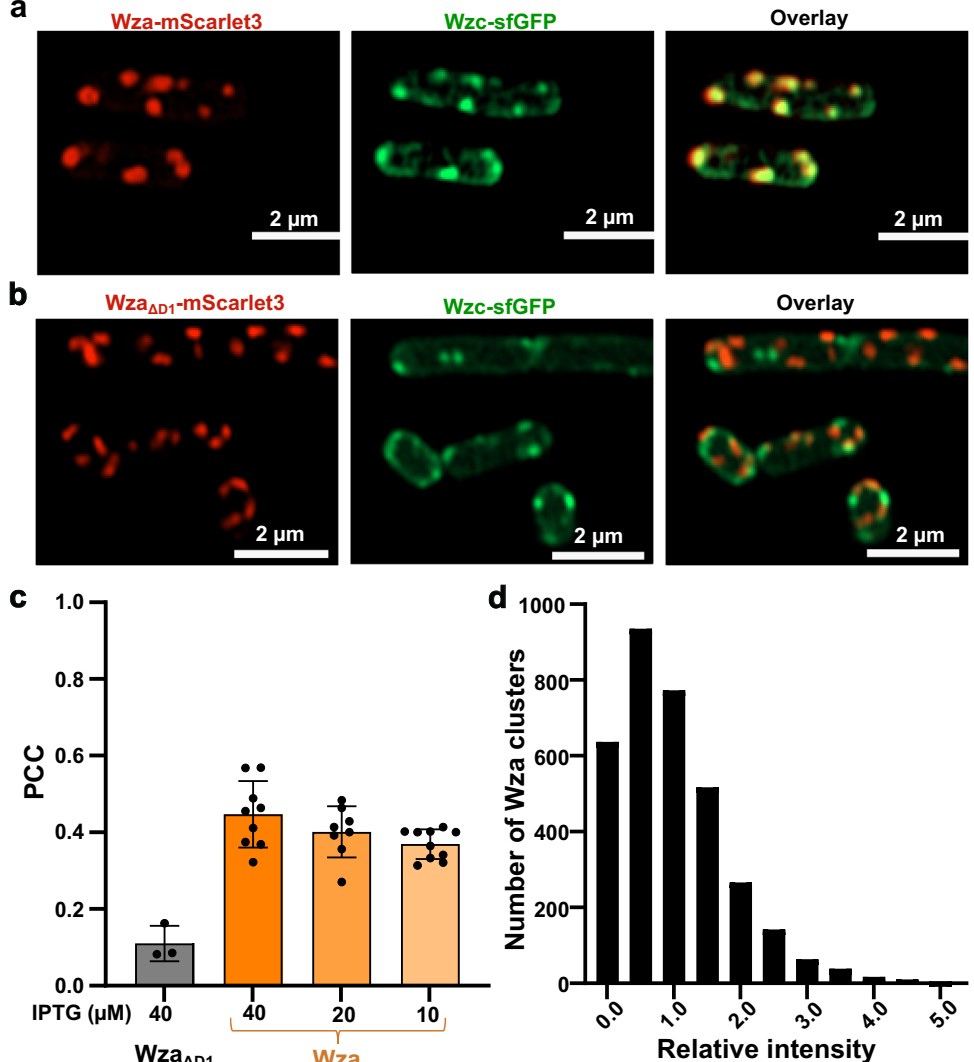

**Fig. 1 | Co-localization of Wza and Wzc in vivo. a, b** Fluorescence confocal microscopy images of live *E. coli* BL21 Star (DE3) cells expressing the *E. coli* K12 *wzabc* operon. C-terminal fusions of Wza-mScarlet3 and Wzc-sfGFP reveal co-localization between Wza and Wzc upon induction with 40 μM IPTG (**a**). Deletion of the Wza D1 domain (**b**) resulted in loss of the typical localization pattern. Scale bar, 2 μm. Wza signals are shown in red, and Wzc signals are shown in green. **c** Colocalization between Wza and Wzc was (**a, b**) quantified by calculating the Pearson's correlation coefficient (PCC). Data are presented as mean values ± s.d. Each dot represents one individual micrograph used for quantification. **d** Wza-mScarlet3 clusters were further imaged using widefield microscopy, diffraction-limited clusters were extracted, and their intensity was quantified.

production. Deletion of the JR domain resulted in a pronounced, yet incomplete, reduction in CPS production. In contrast, a charge-reversal substitution at the channel surface groove (E354R) markedly enhanced CPS production, whereas R374E yielded levels comparable to the wild type, similar to the inner-channel substitution K307E. However, another inner-channel mutation, D301R, completely abolished CPS production, suggesting that CPS translocation occurs within the Wzc channel.

## Cryo-EM structures reveal different conformations of CPS secretion machinery

Magnesium ions ($Mg^{2+}$), essential cofactors for ATP hydrolysis, play a critical role in stabilizing both the enzyme-ADP complex and the transition state during the reaction[15,26,27]. In addition, $Mg^{2+}$ facilitates oligomerization of the Wzc kinase domain[26]. Interestingly, ADP alone binds to the Wzc kinase domain with approximately sixfold greater affinity than the ADP-$Mg^{2+}$ complex[26,27]. As a result, chelation of $Mg^{2+}$ by EDTA could likely capture distinct transient states of the ADP-bound

Wza-Wzc complex. When 10 mM EDTA was added to remove $Mg^{2+}$, 2D classification in fact revealed a significant increase in particles consisting of a single Wzc octamer associated with two Wza translocons compared to the untreated sample (Supplementary Figs. 2b and 3b). Subsequent 3D reconstructions identified five additional conformations (Confs I−V) of the Wza-Wzc complex (Fig. 3 and Supplementary Figs. 3 and 4), highlighting the dynamic nature of the CPS secretion machinery during the secretion process.

The global resolution of the Conf I structure decreased to 3.8 Å compared to that of Conf 0 (C1, Supplementary Fig. 4a). Structurally, Conf I closely resembled the Conf 0 shown in Fig. 2a and, as anticipated, lacked $Mg^{2+}$ in the ADP-binding pocket (Fig. 3a, b and Supplementary Fig. 5a). This absence appeared to trigger a subtle cascade-like motion propagating from the Y-Kinase domain to the HA domain (Supplementary Fig. 5b). Conf II revealed an expanded CPS secretion machinery, approximately 5 Å longer than Conf I (Fig. 3c, d). Structural superposition of the Wza regions from Conf I and Conf II revealed that the Wza translocon remains largely conformationally stable

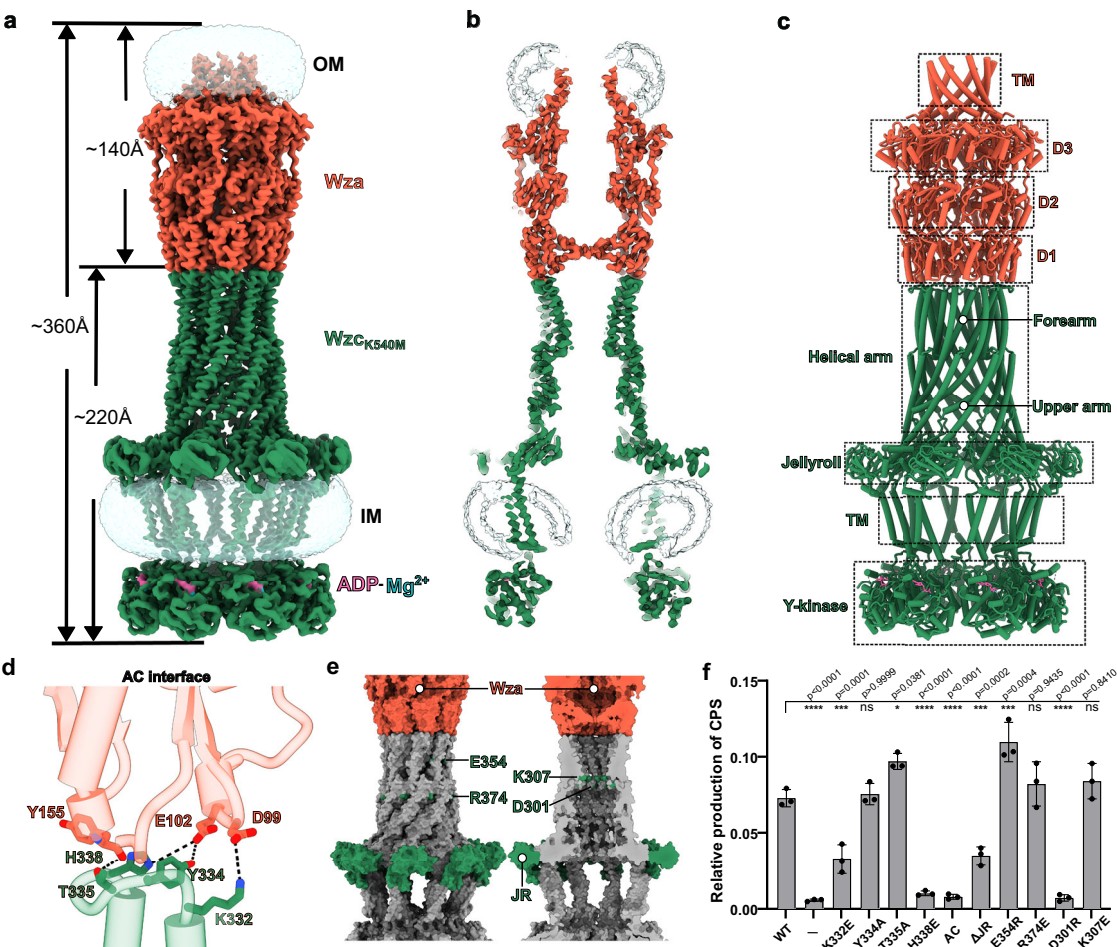

**Fig. 2 | Structural organization of the Wza-Wzc complex. a** Cryo-EM density map of Wza-Wzc$_{K540M}$ complex. Bound ADP-Mg$^{2+}$ is labeled. EM density of DDM micelles is shown in transparent blue. Wza EM density map is shown in tomato red; Wzc EM density map is shown in green. **b** Cross-section of the cryo-EM density map of the Wza-Wzc$_{K540M}$ complex. **c** Atomic model of Wza-Wzc$_{K540M}$ complex with individual subdomains labeled. TM - transmembrane domain; D1, D2, D3 - domains 1, 2, 3; Y-kinase - Tyrosine auto-kinase domain. **d** Loop-mediated interactions at the Wza and Wzc$_{K540M}$ (AC interface). Key residues of the AC interface are indicated with sticks and labeled. **e** Charged residues located at the oligomerization surface interface and at the narrowest region of the periplasmic channel. Key structural elements of Wzc targeted for site-directed mutagenesis are colored in green. **f** Mutational analysis of essential structural elements in Wzc. ΔJR indicates deletion of the JR domain (residues 112–204), replaced with a GGGGS linker; AC refers to the AC interface mutant (Wzc$_{K332E,Y334A,T335A,H338E}$). CPS (colanic acid) production levels are presented as filled black circles, representing the mean ± s.d. from three independent biological replicates. Each circle denotes the average of three technical replicates. Statistical analysis was performed on biological replicate means using one-way ANOVA by the nonparametric Dunnett test in GraphPad Prism. Statistical differences were defined as * $P \leq 0.05$, ** $P \leq 0.01$, *** $P \leq 0.001$, and ns (not significant) $P > 0.05$.

(Supplementary Fig. 6a). Alignment of the structures on Wza showed that Wzc rotates by about 25° and each protomer undergoes an overall displacement of ~20 Å. This coordinated motion suggests that the Wzc HA domain drives the extension of the CPS secretion machinery through a twisting mechanism (Fig. 3e and Supplementary Fig. 6b). Conf III exhibited a partially open periplasmic channel, with one HA forearm positioned inside the incompletely sealed channel (Fig. 3f). In Conf IV, the periplasmic channel was largely open, and the Wza translocon appeared almost detached from the Wzc octamer (Fig. 3g). In this conformation, the Wza translocon remained associated with Wzc via just four HA domains, suggesting that four Wzc protomers are sufficient to maintain Wza association. The remaining four forearms, which were not interacting with Wza, likely became more flexible and were therefore unresolved in the cryo-EM map. Finally, Conf V revealed a configuration in which the HA domains split into two halves (opening angle ~60°), with four protomers in each half associated with a separate Wza translocon (Fig. 3h). This Conf V further confirmed that four accessible HA domains are sufficient to recruit and engage a Wza translocon.

Electrostatic mapping of the Wzc periplasmic domain revealed a large hydrophilic channel with minor changes in charge distribution among Confs I–IV, whereas the channel geometry varied substantially (Supplementary Fig. 6c–f). The balanced distribution of positively (e.g., K307, K326) and negatively charged residues (e.g., D301, E329, E380) within the channel likely contributes to minimizing non-specific electrostatic interactions with the negatively charged CPS polymer. Comparison of Conf I (~17 Å at K307; ~36 Å at E329) with Conf II (~13 Å and ~31 Å) indicates channel narrowing and charge rearrangement at the AC interface, leading to compaction of the forearm region and relaxation of the upper arm (Supplementary Fig. 6c, d). Conf III corresponds to a blocked-channel conformation (Supplementary Fig. 6e), whereas Conf IV displays partial assembly via four forearms, likely representing a transitional intermediate (Supplementary Fig. 6f).

Collectively, these results demonstrate the intrinsic flexibility of the Wzc HA domains, particularly in the forearm regions (Fig. 3i), and these structural transitions support a twisting-driven model in which Wzc rotation dynamically modulates channel architecture to guide CPS export.

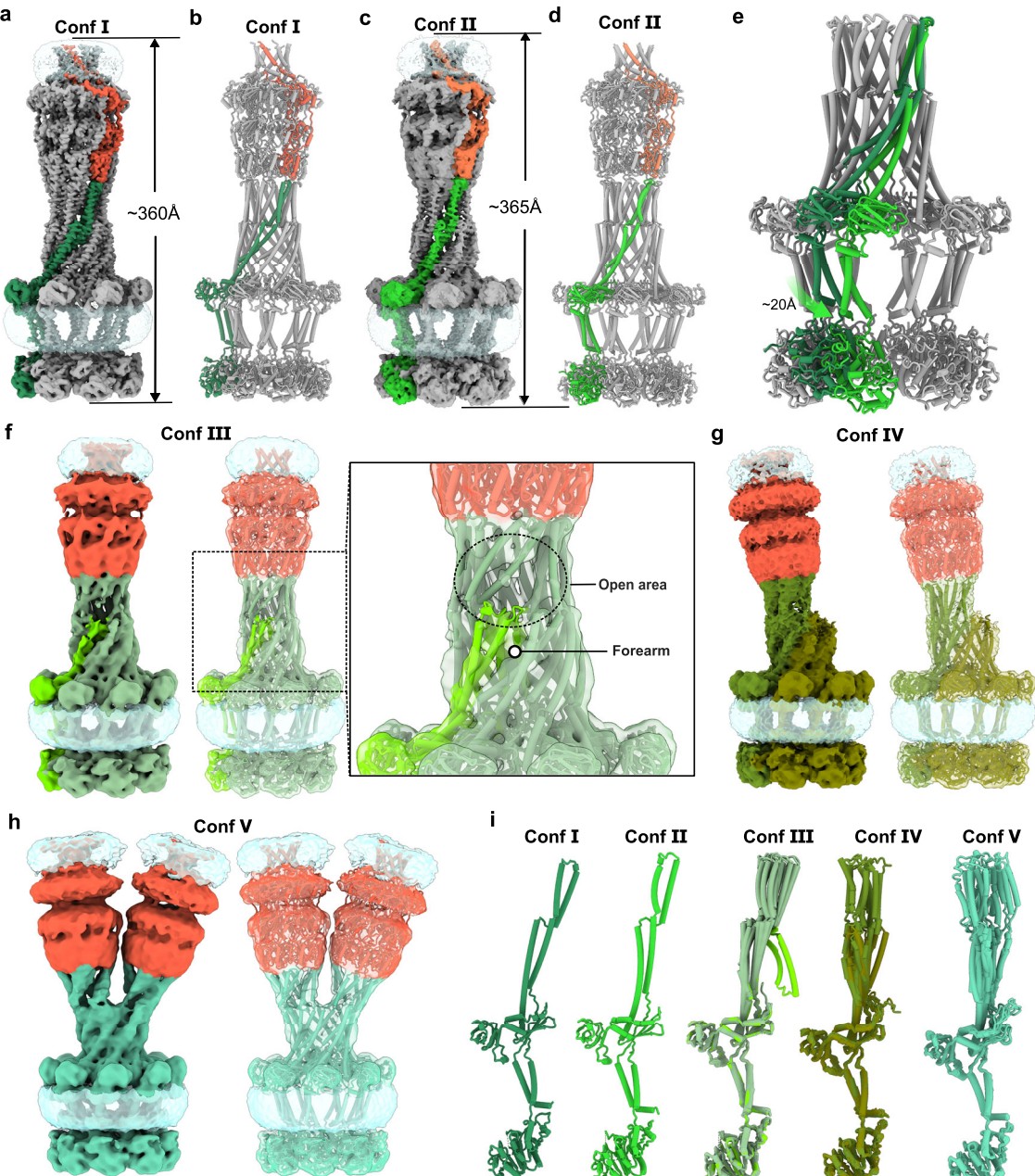

**Fig. 3 | Dynamics of the Wza-Wzc complex. a** Cryo-EM density map of the Wza-Wzc$_{K540M}$ complex in the Conf I state. The density map corresponding to a single Wza protomer is colored in tomato red, while the interacting Wzc protomer is shown in sea green. "Conf" denotes conformation. **b** Structural model of the Wza-Wzc$_{K540M}$ complex in the Conf I state. **c** Cryo-EM density map of Wza-Wzc$_{K540M}$ complex in Conf II state. EM densities corresponding to a Wza protomer and its interacting Wzc protomer are shown in coral red and lime green, respectively. **d** Structural model of the Wza-Wzc$_{K540M}$ complex in the Conf II state. A single Wza protomer and its interacting Wzc protomer are shown in coral and lime green, respectively. The rest of the protomers are shown in gray. **e** Structural superposition of Conf I and Conf II states of the Wza-Wzc$_{K540M}$ complex, aligned on the Wza translocon to highlight conformational differences. The Conf I state Wzc protomer is shown in sea green, and the Conf II state Wzc protomer is colored in lime green. **f** Cryo-EM density map of the Wza-Wzc$_{K540M}$ complex in the Conf III state showing a partially open periplasmic channel with one HA forearm protruding into the channel interior. The EM density map of Wza is colored in tomato red. The EM density map of the Wzc protomer with the forearm lying down is colored in bright green, while that of the others is colored in gray-green. **g** Cryo-EM density map of the Wza-Wzc$_{K540M}$ complex in the Conf IV state, characterized by a large opening at the interface between Wzc and Wza. The EM density map of Wza is colored in tomato red. The EM density map of Wzc protomers with the full HA domain resolved is colored in dark green, and that of the others is colored in yellow-green. **h** Cryo-EM density map of the Wza-Wzc$_{K540M}$ complex in the Conf V state, where a single Wzc octamer is associated with two Wza translocons. The EM density map of Wza is colored in tomato red. The EM density map of Wzc is colored in turquoise green. **i** Comparison of Wzc protomers across Confs I–V. Conf I shows one protomer from the Conf I state; Conf II shows one protomer from the Conf II state; Conf III-V shows a superposition of all 8 protomers from each state. Wzc protomers are colored according to the same scheme used in panels (**a**–**h**).

## Insights into the Wza-Wzc assembly

The Wza-Wzc complex forms an octameric trans-envelope structure essential for CPS secretion in most Gram-negative bacteria. This large, dynamic assembly spans both membranes, with the observed conformational flexibility of the Wzc HA domain playing a pivotal role in coordinating CPS polymerization and export through the periplasm. To elucidate how Wza and Wzc initiate assembly of this nanomachine, we determined the cryo-EM structures of the isolated Wza translocon

at a resolution of 3.0 Å (C1) and 2.7 Å (C8) (Supplementary Fig. 7), along with the structures of the isolated Wzc$_{K540M}$ octamer at 3.4 Å (C1) and 3.2 Å (C4) resolution (Supplementary Fig. 8). In addition, some micrographs of *E. coli* K12 Wza revealed 2D crystalline arrangements (Supplementary Fig. 7a), consistent with those observed for *E. coli* K30 Wza[22].

The Wza translocon maintained the same conformation in its isolated octameric form as it did when complexed with Wzc (Fig. 4a–c), indicating high structural rigidity and stability across different functional states of the complex. In contrast, the isolated Wzc$_{K540M}$ octamer exhibited marked conformational diversity, existing in two distinct protomeric states within the octamer. In Conformation i (Conf i), four protomers assembled into a V-shaped channel, with the forearms of the HA domains tightly aligned. In Conformation ii (Conf ii) the upper arms of four protomers rotated outward, leaving the forearms more flexible and unresolved in the structure (Fig. 4d–g). This suggests that in Conf ii, the forearms become exposed and dynamic, unlike their shielded position in Conf i (Fig. 4f,-g), allowing Wzc to engage and recruit the Wza translocon.

For complete translocation channel formation with Wza, Wzc must undergo a rotational conformational shift toward the outer membrane, extending all forearms to establish a sealed translocation pathway (Supplementary Fig. 9a, b). Notably, the periplasmic domain of Wzc (Wzc$_{Peri}$), comprising the HA- and JR-domains, alone was unable to form a complex with Wza (Supplementary Fig. 9c), supporting a model in which oligomerization of the cytoplasmic Y-kinase domain drives assembly by exposing HA forearms necessary for Wza engagement. This mechanism supports previous findings[13,20,24] linking Y-kinase domain-driven transitions to Wza-Wzc complex formation, regulated by the phosphorylation state of the Y-tail. To dissect this relationship, we analyzed the phosphorylation state of the recombinantly produced wild-type Wzc by mass spectrometry. Wild-type Wzc existed as a lower-order oligomeric form in vitro (Supplementary Fig. 9d). Mass spectrometry analysis demonstrated that while up to four tyrosines can undergo simultaneous phosphorylation, with the peptides bearing one (1pY) or two (2pY) phosphorylated tyrosines predominated (Supplementary Fig. 9e). The single phosphorylation Y705 was found most abundant, followed by Y708 and Y715. Association analyses indicated higher order phosphorylations have a preference to involve the sites Y710, Y713 and Y715 (Supplementary Fig. 9f). In total, 43 different phosphorylation site combinations (with phosphorylation probabilities above 0.85) were detected, suggesting a redundant phosphorylation pattern along dynamic assembly (Supplementary Table 1). This redundancy aligns with the observation that no single phosphorylation site is indispensable for CPS secretion[28].

Compared to the previously reported *E. coli* K30 Wzc$_{K540M}$ structure with a wide open inner membrane cavity (3.5 nm in diameter)[13,29], the *E. coli* K12 Wzc$_{K540M}$ exhibited a C4 symmetric conformation, with the inner membrane cavity sealed by the forearms in Conf i (Fig. 4h and Supplementary Fig. 8g, h). The closed configuration of the Wzc octamer likely represents its native in vivo state, ensuring that the central cavity remains impermeable to ions and small molecules before association with Wza, thereby maintaining membrane integrity and electrochemical balance. Superimpositions of the Wzc protomers from the *E. coli* K30 and K12 strains indicated that the JR domain might move vertically relative to the inner membrane (Fig. 4i). In addition, the presence of a hydrophobic helix (residues 118–129) within the JR domain (Fig. 4i), located near the inner membrane, suggests transient lateral membrane association that may be disrupted during conformational transitions.

**Wzc JR domain enables CPS repeat unit recognition and loading**
Although we resolved the Wza-Wzc complex required for CPS polymerization and export, how CPS is loaded into this machinery and in particular, how Wzc and Wzy interact during CPS polymerization

remain unclear. Recent structural and biochemical studies of the homologous WzzE (co-polymerase)-WzyE (polymerase) complex, involved in polymerization of the enterobacterial common antigen, suggest the formation of a polysaccharide polymerization platform with an 8:1 stoichiometry between co-polymerase and polymerase[10]. Weckener et al. hypothesized that this structural arrangement might also apply to its homologous CPS polymerization system where Wzc functions as the co-polymerase[10]. To further test this hypothesis, we used AlphaFold 3[30] to predict a putative Wzc-Wzy (ratio 8:1) complex structure (Fig. 5a and Supplementary Fig. 10). The AlphaFold prediction suggests that CPS polymerization may occur on an inner membrane platform resembling the WzzE–WzyE complex[10], characterized by a moderate confidence (ipTM score: 0.56).

This proposed positioning of Wzy within the inner membrane cavity of Wzc (Fig. 5a) raises the question of how CPS repeat units are subsequently delivered to the polymerization platform after their translocation across the inner membrane by the flippase Wzx. Strikingly, the JR domain of Wzc, with a yet unknown function, is situated near the periplasmic side of the inner membrane. The jellyroll fold is a structural motif commonly found in various carbohydrate-binding proteins, including L-type lectins, carbohydrate-binding modules (CBMs), and members of the glycoside hydrolase family 16 (GH16)[31–33]. Positioned near both the membrane and the core of the proposed Wzc-Wzy polymerization platform, this domain is well-placed to facilitate the transfer of membrane-anchored CPS repeat units from the flippase Wzx to the polymerase Wzy. Importantly, deletion of the JR-domain nearly abolished CPS production (Fig. 2f), indicating its critical role in coupling CPS to the secretion machinery.

Based on these structural analyses, we propose that the JR domain acts as a recognition module for CPS repeat units, aiding in their transfer from Wzx to the CPS polymerization platform. Superposition of the JR domain of Wzc$_{K540M}$ with the crystal structure of CBM29-2 bound to hexasaccharide mannohexaose[32] shows a potential CPS repeat unit binding site in the JR domain. The location of the mannohexaose in this superposition suggests an analogously bound membrane-anchored CPS repeat unit interacting in a similar fashion with the JR domain of Wzc (Fig. 5b). The positioning of the CPS repeat unit within the Wzc octamer further reveals a potential substrate entry site towards the Wzc-Wzy polymerization platform. This site is formed by the inner membrane plane along with the TM and JR domains of two neighboring Wzc protomers and appears to be sufficiently large (~15 Å high and ~24 Å wide) for CPS repeat units entry (Fig. 5c). We therefore propose that CPS repeat units enter the polymerization platform through this site, where the JR domain captures and directs them into the inner membrane cavity for polymerization by Wzy. Polymer growth then proceeds toward the Wza translocon, guided by the positively charged HA channel of Wzc (Fig. 5c).

To further assess whether the JR domain is capable of recognizing CPS, we conducted a glycan array screen using both the isolated JR domain (Wzc$_{JR}$) and the entire periplasmic region (Wzc$_{Peri}$) at three different protein concentrations (Fig. 5d and Supplementary Fig. 11). Glycan array profiling revealed 13 glycans (Supplementary Table 4) consistently bound to both Wzc$_{JR}$ and Wzc$_{Peri}$ (Supplementary Fig. 11). These glycans are notably enriched in N-acetylglucosamine (GlcNAc), galactose (Gal), and mannose (Man) residues, indicating a recognition preference centered around GlcNAc and Gal. The recurring presence of these sugars suggests that the GlcNAc-Gal linkage serves as a key recognition scaffold. Structurally, the bound glycans closely resemble the repeating-unit composition of colanic acid, which includes Gal-GlcNAc-Fuc-GlcA. Furthermore, consistent binding to aminoglycosides and acidic sugars points to an electrostatic contribution to glycan recognition. Altogether, these results suggest a GlcNAc-Gal-based carbohydrate recognition motif reinforced by charge-mediated interactions, supporting that the Wzc JR domain contributes to CPS assembly.

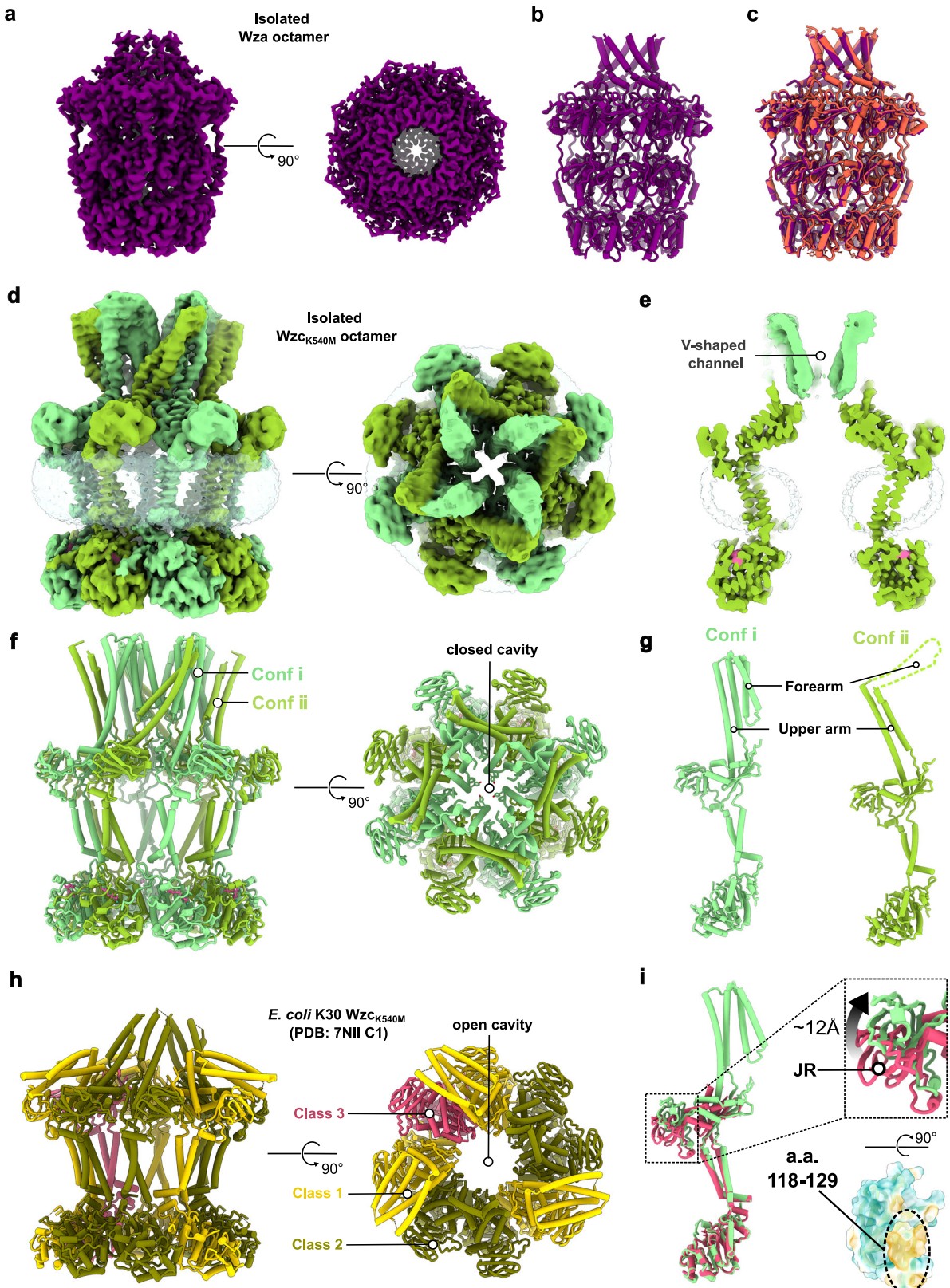

## Discussion

Synthesis and transport of CPS via the Wzx/Wzy-dependent pathway plays a fundamental role in the virulence, environmental adaptability, and immune evasion strategies of most Gram-positive and Gram-negative bacteria[34]. In this study, we present the complete structure of Gram-negative bacterial CPS secretion machinery formed by the *E. coli*

K12 Wza-Wzc complex. The structure, which is an octameric hetero-dimer, reveals a 360 Å long trans-envelope channel that spans the inner and outer membranes. Multiple structural snapshots of the Wza-Wzc complex, along with the isolated Wza and Wzc octamers, offer important insights into the assembly and disassembly dynamics of the CPS secretion machinery.

**Fig. 4 | Cryo-EM structures of the isolated Wza translocon and Wzc octamer.**
**a** Cryo-EM density map of the Wza translocon resolved with C8 symmetry. The EM map is colored in purple. **b** Structural model of the Wza translocon. The model is colored in purple. **c** Structural comparison of Wza before (purple) and after (tomato red) complex formation with Wzc, illustrating its conformational stability. **d** Cryo-EM map of the isolated *E. coli* K12 Wzc$_{K504M}$ octamer obtained by C1 reconstruction. The EM density map of Conf i protomers is colored light green. The EM density map of Conf ii protomers is colored in yellow-green. **e** Cross-sectional view showing the inner membrane cavity of *E. coli* K12 Wzc$_{K504M}$ sealed by the HA domains in Conf i of Wzc$_{K504M}$. **f, g** Structural model of *E. coli* K12 Wzc$_{K540M}$ octamer generated based on the C1-reconstructed EM density map. Conf i protomers are colored light green; Conf ii protomers are yellow green. Only the HA forearm of the Conf ii protomer is resolved. **h** Model of *E. coli* K30 Wzc$_{K540M}$ (C1 symmetry; PDB: 7NII)[13]. The Wzc protomers corresponding to classes 1–3 are shown in gold, dark green, and red, respectively. **i** Superimposition of *E. coli* K12 Conf i Wzc$_{K540M}$ and *E. coli* K30 Wzc$_{K540M}$ (class 3) protomers reveals vertical displacements of the JR domain. The hydrophobic helix (residues 118–129) within the JR domain lies near the inner membrane in the class 3 conformation, suggesting membrane association. Upward movement of the JR domain during conformational transitions (class 3 to Conf i) likely facilitates its dissociation from the membrane, potentially contributing to the CPS translocation. a.a. (amino acids).

The periplasmic width of *E. coli* is approximately 330 Å, though this spacing is not uniform and can vary between ~210–440 Å depending on envelope stress and the presence of trans-envelope assemblies[17]. Within this physiological range, the ~250 Å span of the Wza-Wzc complex is structurally plausible and comparable to other envelope-spanning machines, including the type III secretion system injectisome (~250 Å)[35], type IV secretion system (~210 Å)[36] and the AcrAB-TolC efflux pump (~210 Å)[37]. Such complexes are thought to locally constrict the periplasm by bending and compressing the membranes, thereby establishing a continuous conduit across the envelope. This local narrowing may enhance transport efficiency by reducing diffusion losses and providing a protected passage for substrates. In addition, the Wza-Wzc complex is primarily stabilized by electrostatic interactions, suggesting that fluctuations in periplasmic ionic strength could significantly impact its assembly dynamics. Although these parameters were not directly assessed here, they represent an important factor that may fine-tune the efficiency and regulation of CPS export.

While the Wza translocon maintains a strikingly stable conformation during these transitions, the Wzc co-polymerase exhibits substantial conformational rearrangements, particularly within its HA domain. This domain transitions from a rather compact form to an extended state, allowing it to reach and interact with Wza for complex formation. Resolved structures of the Wza-Wzc complex also reveal distinct conformational states, emphasizing the intrinsic flexibility and dynamic coordination required for CPS transport. The HA domain's structural variability highlights its central role in modulating CPS polymerization and export. Notably, in the absence of Mg$^{2+}$, conformational changes are initiated, propagating from the cytoplasmic Y-kinase domain to the HA domain. This results in a dramatic transition from a stable, continuous secretion channel to a widely open conformation, revealing a potential mechanical basis for CPS extrusion and subsequent disassembly of the CPS secretion machinery.

Multiple conformational changes via the HA domain of Wzc provide molecular insights into how the Wza-Wzc machinery mediates CPS secretion through coordinated structural changes. Wza-Wzc forms a broad aqueous channel that extends across the periplasm, enabling the translocation of a branched, flexible, high molecular weight CPS polymer[38]. The twisting-driven motion of Wzc and the accompanying modulation of the periplasmic channel from Conf I to Conf II reveal a coordinated mechanism in which mechanical rotation and pore-size adjustments act together to facilitate CPS translocation. Importantly, this elongation from Conf I to Conf II likely generates a mechanical force sufficient to extract the final CPS repeat unit from the inner membrane, thereby aiding its dissociation from Wzy. By coupling torsional motion to channel constriction, Wzc may act as a molecular winch that pulls out CPS polymer from the polymerization platform. Conf III likely corresponds to the loading state of CPS from Wzc to Wza, where the forearms occupy the channel lumen, closing the inner membrane cavity and halting additional CPS synthesis. The side-open channel of Conf IV exposes the inner membrane cavity, probably accommodating the growing CPS polymer. Once the polymer reaches full length, the channel completes its assembly to facilitate CPS export. An intriguing aspect of our study is the identification of a Wzc octamer associated with two Wza translocons (Conf V). This configuration may represent an in vitro complex formed during isolation. Alternatively, it could be due to an EDTA-induced dissociation and subsequent artifactual re-association of the Wza-Wzc complex. Nevertheless, the observed arrangement suggests that four helical arms of the Wzc octamer are sufficient to recruit a Wza translocon. Given the known tendency of Wza to form higher-order clusters in vivo (Fig. 1d), Conf V may correspond to a transient intermediate that facilitates dynamic modulation between Wzc and Wza, thereby enhancing CPS secretion efficiency. However, further in vivo investigations and high-resolution imaging approaches such as cryo-ET and MINFLUX nanoscopy will be important to determine whether Conf V naturally occurs in bacterial CPS transport systems.

While our cryo-EM and biochemical data offer insight into the Wzx/Wzy-dependent synthesis pathway, the broader evolutionary context and mechanistic distinctions among bacterial polysaccharide transport pathways merit further consideration. In the ABC-transporter-dependent pathway, CPS is synthesized within the cytoplasm and translocated across the inner membrane by the ABC transporter complex[4]. A recent cryo-EM structure of KpsMT bound to its periplasmic adapter KpsE revealed a potential secretion pathway through the inner membrane[39]. Moreover, AlphaFold modeling of the KpsMT-KpsE in complex with outer membrane translocon KpsD bears striking structural resemblance to the Wza-Wzc machinery characterized in our study[39]. This convergence implies a common evolutionary mechanism across Gram-negative species to develop large, contiguous secretion supercomplexes that enable the efficient and tightly regulated export of CPS. Further comparative and phylogenetic analyses will be instrumental in unraveling whether these systems arose via convergent or divergent evolutionary trajectories.

Unlike the relatively simple O-antigens, CPSs often display large chemical and structural diversity, frequently incorporating a broad repertoire of monosaccharides, amino sugars, and occasionally non-carbohydrate moieties such as pyruvate or acetyl groups[1,40]. This structural diversity enables bacteria to tailor their capsules to diverse ecological niches and host immune environments[1,40]. Intriguingly, this complexity is mirrored in the architecture of their co-polymerase. Two main mechanisms have been proposed for the function of polysaccharide co-polymerase 1 (PCP-1) Wzz, either serving as a scaffold that coordinates chain elongation among multiple Wzy molecules or acting as a lumen within which polymerization occurs[10,41]. Unlike PCP-1 Wzz, PCP-2a Wzc contains an additional cytoplasmic Y-kinase domain and a JR domain (Supplementary Fig. 12)[13,29]. Although JR deletion does not abolish CPS production, it substantially reduces yield, indicating a supportive rather than essential role. Thus, we propose the JR domain functions as a carbohydrate-recognition module that facilitates the capture and delivery of CPS repeat units to the polymerization platform. The proposed substrate entry site, formed by adjacent Wzc protomers, provides a mechanistic explanation for how CPS repeat units are funneled into the inner membrane cavity for elongation. Wzc

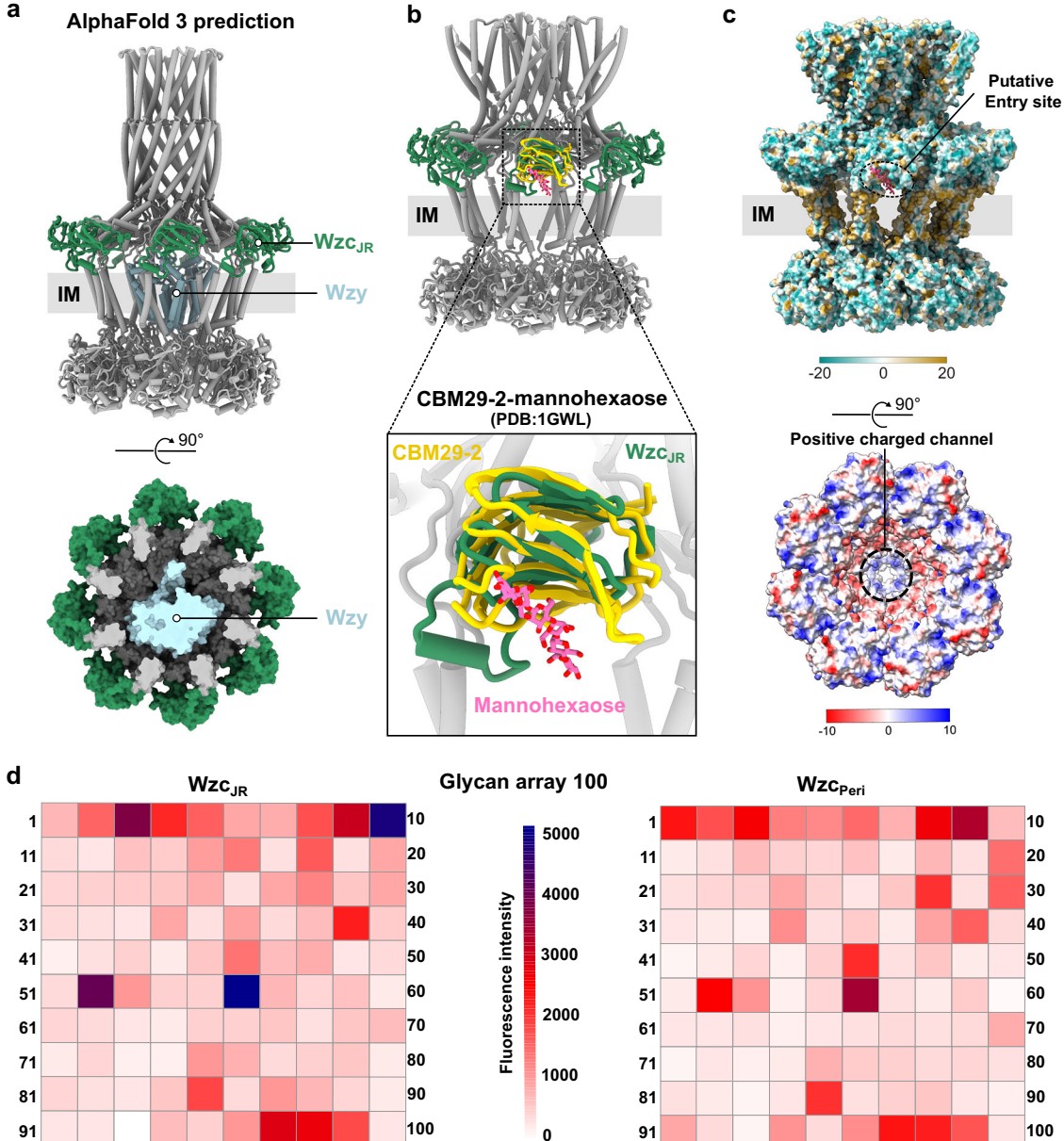

**Fig. 5 | The CPS polymerization platform Wzc-Wzy with JR-directed CPS loading. a** AlphaFold3 model of the Wzc-Wzy complex. Side (top panel) and bottom (bottom panel) views of the predicted structure reveal that the Wzc inner membrane cavity is sufficiently spacious to accommodate a single Wzy molecule. The co-polymerase Wzc JR domain (WzcJR) is colored in sea green, and the polymerase Wzy is colored in light blue. **b** Superimposition of the WzcK540M structure with the mannohexaose bound structure of CBM29-2[32](PDB: 1GWL, CBM29-2 in yellow; mannohexaose in pink). **c** Surface properties of the WzcK540M structure reveal a putative CPS repeat unit entry site and a positively charged CPS polymerization channel. Surface hydrophobicity highlights the proximity of the superimposed mannohexaose to the putative CPS repeat unit entry site, formed by JR, TM, and the inner membrane plane (Top panel). Surface electrostatic potential of the WzcK540M octamer reveals a positively charged channel likely guiding CPS polymerization towards the Wza translocon. The transmembrane region is labeled as IM. JR domain is labeled as WzcJR. **d** Glycan binding profiles of *E. coli* K12 WzcJR and WzcPeri (Wzc periplasmic domain) using Glycan Array 100 (Raybiotech). Cys3-conjugated Streptavidin are used to detect the binding ability of biotinylated WzcJR and WzcPeri. The relative fluorescence intensity of WzcJR (25 μg/ml) and WzcPeri (50 μg/ml) binding to each glycan printed on the glass slide provided from the Glycan Array 100 kit is presented in a heatmap format. The scale bar is from 0 to 5000.

can bind a wide range of glycans, providing the molecular basis for CPS diversity and specificity. The functional flexibility of the JR domain likely underlies the ability of the Wza-Wzc system to accommodate various CPS types, as exemplified by its capacity to mediate the export of both colanic acid and K30 CPS via *E. coli* K12 Wza-Wzc complex[7]. The JR domain exhibited affinity toward specific carbohydrate motifs based on the glycan microarray data, but failed to interact strongly with the intact colanic acid polymer, as evidenced by the negative results from our cosedimentation assay using colanic acid–producing

*E. coli* JM109 (DE3) with purified sfGFP-WzcJR (Supplementary Fig. 11c). This suggests that the structural complexity and higher-order organization of the polysaccharide might critically influence molecular recognition. These findings imply that the binding epitopes identified in vitro may not be readily accessible or appropriately presented within the native polymer matrix. Future studies will be required to quantitatively characterize the binding affinity of the JR domain toward both individual colanic acid repeat units and the fully assembled polymer.

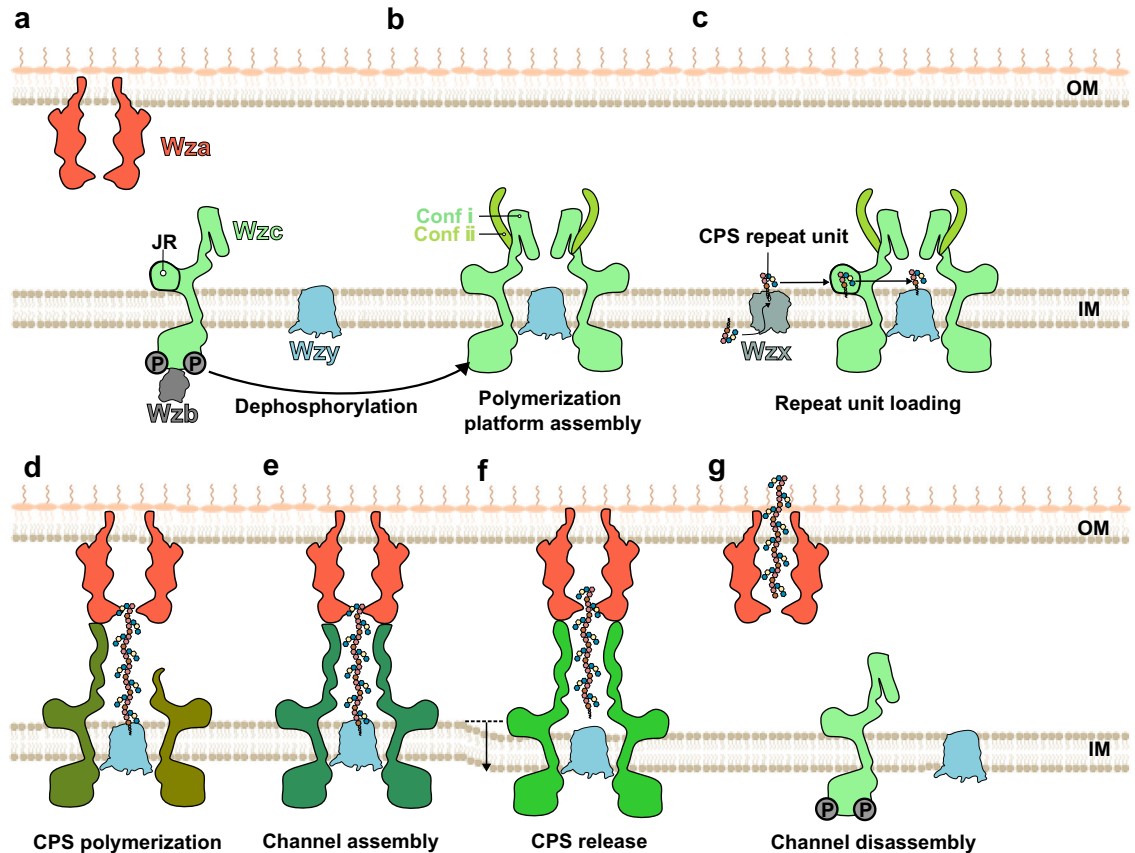

**Fig. 6 | Proposed model of CPS polymerization and secretion. a** Preassembly state of the CPS secretion machinery. The Wza translocon (in tomato red) is pre-assembled as an octamer in the outer membrane (OM). Phosphorylated Wzc (in green) remains monomeric within the inner membrane (IM), with its HA domain adopting a conformation that prevents interaction with Wza. The cytoplasmic phosphatase Wzb can access and dephosphorylate phosphorylated Wzc. **b** Initiation of CPS polymerization platform assembly. Dephosphorylation by Wzb triggers Wzc oligomerization, likely around the CPS polymerase Wzy (in light blue), leading to the formation of a functional polymerization platform. **c** CPS repeat-unit recognition and loading. CPS repeat units, flipped from the cytoplasmic to the periplasmic side by Wzx (in gray), are recognized by the JR domain and subsequently delivered to Wzy within the inner-membrane cavity of Wzc for polymerization. **d** CPS polymerization. Polymer elongation along the HA domain promotes exposure of its forearm conformation, enabling recruitment of the preassembled Wza translocon. **e** Assembly of the CPS secretion machinery. Once the CPS chain reaches maturity, the HA domains open, forming a continuous secretion channel with Wza translocon that spans the entire periplasm. **f** CPS release. Twisting of the HA domains extends the secretion channel and may provide the mechanical force required to release the CPS polymer from the inner membrane into the Wza translocon. **g** Disassembly of the CPS secretion machinery. The mature CPS is exported across the outer membrane via Wza, followed by disassembly of the secretion machinery through Wzc auto-phosphorylation.

Our data further support a phosphorylation-dependent mechanism that regulates Wzc oligomerization[14,20,24]. Phosphorylation analysis of wild type Wzc indicates that the Y-tail undergoes multiple phosphorylation events, with Y710, Y713, and Y715 being the most frequently modified among higher order phosphorylations, whereas single phosphorylation is mainly associated with Y705. These findings broaden our understanding on the involvement of transient phosphorylation patterns during the dynamic assembly of Wzc. Dephosphorylation of the C-terminal Y-tail triggers the Y-kinase domain octamerization[14,20,24], which further promotes the rearrangement of HA domains to engage the Wza translocon. Wzz, lacking the Y-kinase domain, instead can form an octameric assembly directly via its HA domain[24,42] (Supplementary Fig. 12). Although the HA domain architecture is broadly conserved between Wzc and Wzz, Wzc's HA is significantly extended[29], suggesting functional divergence. This elongation, along with the presence of the additional regulatory domains, such as JR domain, underscores the specialized adaptations of Wzc to support CPS assembly and export via the Wza translocon, distinguishing it from Wzz's role in lipopolysaccharide assembly and secretion.

Our findings support a proposed model for CPS polymerization and export (Fig. 6), which, while hypothetical, provides a conceptual framework for understanding the underlying processes. According to this model, following the SecYEG-SPII maturation of Wza[43], the protein assembles into a stable octameric translocon in the outer membrane, independently of other components[19]. In contrast, Wzc predominantly exists in a monomeric or heterogeneous oligomeric states when phosphorylated[13]. Dephosphorylation of the tyrosine residues in the C-terminal Y-tail by the cognate phosphatase Wzb triggers the assembly of Wzc into a stable octamer likely surrounding the polymerase Wzy[10,34]. Conf i HA domains form a polymerization platform with Wzy, directing CPS chain elongation toward the Wza translocon. Meanwhile, the flexible Conf ii HA domains facilitate recruitment of the Wza translocon. Positioned at the periplasmic face of the inner membrane, the JR domain acts as a glycan receptor, capturing CPS repeat units flipped by Wzx and delivering them to Wzy for polymerization. As polymer elongation advances, conformational rearrangements within the HA domain of Wzc promote exposure of its forearm conformation, enabling the continuous secretion conduit assembly, linking the polymerization site to the extracellular environment. The progressive alignment of the Wza-Wzc complex likely ensures that elongation and export proceed in synchrony, preventing periplasmic accumulation of intermediate-length polymers. Upon twisting, the HA domain drives an approximately 25° rotation of the Wzc complex within the inner

membrane. This coordinated motion most likely exerts mechanical stress through the HA domain, promoting the release of the CPS polymer from Wzy. Upon completion of translocation, the CPS secretion machinery dissociates through Wzc autophosphorylation, effectively resetting the machinery for a new cycle of CPS biosynthesis and export. This model offers a basis for future experimental investigation into the mechanisms governing CPS biosynthesis and transport.

Together, our structural and biochemical data reveal a highly dynamic CPS secretion machinery, in which coordinated domain rearrangements and glycan recognition converge to ensure efficient CPS biosynthesis and export. This work advances our understanding of the bacterial CPS secretion machinery and may inform the development of antimicrobial strategies by specifically targeting polysaccharide transport systems involved in bacterial virulence.

## Methods

### Constructs, bacteria, and growth conditions

The strains and constructs used in this study are listed in Supplementary Table 2. All bacteria strains used for cloning and expression were grown in LB medium. For the colanic acid secretion assay, the strains were grown in minimal M9 medium supplemented with glucose as a carbon source. *E. coli* JM109 (DE3) Δ*wzabc* strain was generated from *E. coli* JM109 (DE3) by replacing the *wzabc* operon with a chloramphenicol resistance cassette[44]. All expression constructs generated via Gibson assembly or QuikChange PCR and JM109 (DE3) variant strains were verified by DNA sequencing.

### Fluorescence microscopy

*E. coli* BL21 (DE3) strain carrying pET21-Wza-mScarlet3-Wzb-Wzc-sfGFP was grown at 37 °C in LB medium supplemented with 100 mg/ml Ampicillin. The overnight preculture was used to inoculate LB medium supplemented with 100 mg/ml Ampicillin at 37 °C. When the $OD_{600}$ reached 0.6, protein expression was induced with varying concentrations of IPTG (10 μM, 20 μM, and 40 μM) and the cultures were incubated overnight at 19 °C. The cells were then collected and washed twice with 1 x PBS. The $OD_{600}$ was measured and adjusted to be 0.3 with 1 x PBS. Super-resolution imaging was performed using an LSM980 Airyscan2 confocal microscope (Zeiss). Images were processed using the Zeiss Zen software. Widefield fluorescence imaging for cluster quantification was performed using a Nikon Ti Eclipse N-STORM microscope equipped with laser lines at 488 nm (Sapphire, Coherent) and 561 nm (Sapphire, Coherent). The sample was observed through a Nikon Apo TIRF 100x Oil DIC N2 NA 1.49 objective, and emitted light was detected on an Andor iXon3 DU-897 EMCCD camera (Oxford Instruments).

### Colocalization analysis and protein cluster quantification

Multicolor fluorescence microscopy images taken on the LSM980 Airyscan2 were processed using Fiji[45]. Colocalization was quantified using the Fiji plugin EzColocalization[46]. The whole field of view was analyzed following a manual intensity segmentation to specifically focus on the clustered fraction of Wza-mScarlet3. Diffraction-limited clusters of Wza-mScarlet3 were imaged using a Nikon Ti Eclipse N-STORM microscope. The clusters were identified and quantified using the ThunderSTORM[47] localization plugin for Fiji.

### Protein expression and purification

The *wzabc* operon was amplified from *E. coli* K12 genomic DNA using standard PCR techniques using the primers Wzabc_Fw and Wzabc_Rv (Supplementary Table 2). The vector backbone, excluding the ribosome binding site (RBS), was amplified from the pET21 vector using pET21-V_Fw, and pET21_V_Rv. The construct pET21-Wza-Wzb-Wzc was assembled via Gibson assembly. Subsequently, a Strep-tag was introduced at the C-terminus of Wza using site-directed PCR mutagenesis. The point mutations and pET21-Wzb-Wzc-strep were individually

generated using the same mutagenesis method. The fluorescence fusion construct pET21-Wza-mScarlet3-Wzb-Wzc-sfGFP was generated using Gibson assembly.

For large-scale protein purifications, *E. coli* BL21 Star (DE3) star was transformed with the pET21-Wza-strep-Wzb-Wzc$_{K540M}$ plasmid and grown overnight at 37 °C. An overnight preculture of this transformed strain was used to inoculate 4 liters of LB broth supplemented with 100 μg/ml Ampicillin and grown until an $OD_{600}$ = 0.8 was reached. Protein expression was induced with 0.1 mM IPTG at 18 °C for 16 h. Cells were harvested by centrifugation at $5000 \times g$ for 15 min and stored at −20 °C. Frozen cells were thawed and suspended in 30 ml lysis buffer (50 mM Tris-HCl pH 8.0, 300 mM NaCl, 20% sucrose, 2 mM $MgCl_2$) supplemented with one tablet cOmplete™ EDTA-free Protease-Inhibitor-Cocktail (Roche). Cells were lysed by six passes through an Emulsiflex-C3 cell disrupter (Avestin). 20% n-dodecyl-ß-D-maltoside (DDM) was directly added to the lysate to a final concentration of 1% and stirred at 4 °C for 2 h. The lysate was ultra-centrifuged at 150,000 x $g$ for 40 min to remove insoluble materials. The supernatant was then incubated at 4 °C overnight with 2 ml Strep-tactin XT resin. The resin was subsequently washed inside a gravity column with 100 ml washing buffer (50 mM Tris-HCl, pH 8.0, 150 mM NaCl, 20% sucrose, 6 mM $MgCl_2$, 0.05% DDM). Bound protein was eluted as 1 ml fractions with 10 ml total elution buffer (50 mM Tris-HCl, pH 8.0, 150 mM NaCl, 20% sucrose, 6 mM $MgCl_2$, 50 mM biotin, 0.05% DDM). The protein concentration in the elution fractions was measured by absorbance at 280 nm. Protein-containing fractions were pooled and concentrated using Amicon Ultra 0.5 ml 100 kDa spin concentrators. The concentrated sample was further purified by size-exclusion chromatography using a Superose 6 Increase 10/300 GL column equilibrated with SEC buffer (50 mM Tris-HCl 150 mM NaCl 0.02 % DDM, 5% glycerol, 2 mM $MgCl_2$, 2 mM TCEP). Peak fractions corresponding to the Wza or Wza-Wzc$_{K540M}$ complex were collected separately. The Wza complex was concentrated into 2 mg/ml, and the Wza-Wzc$_{K540M}$ complex was concentrated to 6 mg/ml. Samples were flash frozen in liquid nitrogen (LN) and stored at −80 °C for cryo-EM.

The isolated Wzc$_{K540M}$ octamer was purified from *E. coli* BL21 Star (DE3) transformed with pET21-Wzb-Wzc$_{K540M}$-strep, following the same procedure described above. After SEC, the peak fraction of Wzc$_{K540M}$ was concentrated to 2.5 mg/ml, flash-frozen in LN, and stored at −80 °C for cryo-EM.

### Structural analysis by cryo-EM

Samples were vitrified on Quantifoil 200-mesh 1.2/1.3 Gold/Copper grids. Briefly, 3.5 μl of the sample was applied onto glow-discharged grids (PELCO easiGlow™ glow discharger, Ted Pella, Inc; glow discharge conditions, -15 mA for 60 s). The grids were plunge-frozen in liquid ethane using a Vitrobot Mark IV (blot time, 4 s; blot force, −9; temperature, 6 °C; humidity, 100%).

The vitrified specimens were imaged on a Glacios TEM operating at 200 kV, equipped with a Selectris energy filter set to 10 eV, and a Falcon 4i camera running in counting mode and using Thermo Fisher Scientific EPU software. Automated data acquisition was performed using EPU, and movies were recorded with a nominal magnification of ×130,000, corresponding to a pixel size of 0.91 Å at the specimen level. Movies were recorded with a cumulative electron dose of 40 $e^-/Å^2$ with a defocus ranging from −2.0 μm to −0.6 μm.

All data sets were processed in cryoSPARC v4, and cryo-EM density maps were interpreted using ChimeraX 1.9. AlphaFold models were used as initial models, which were iteratively improved with ISOLDE[48] and PHENIX[49]. Further details are provided in Supplementary Table 3.1 and 3.2.

### Glycan array screening

Glycan array screening was conducted using the Glycan Array 100 kit purchased from RayBioTech (Catalog #: GA-Glycan-100-1; Norcross,

GA, USA). This Glycan-100 array was used to assess carbohydrate binding abilities to 100 described glycan structures of $Wzc_{JR}$ and $Wzc_{Peri}$.

Firstly, the biotinylated $Wzc_{JR}$ and $Wzc_{Peri}$ samples were prepared as follows: *E. coli* BL21 Star (DE3) star cells were co-transformed with pACYC-BirA and pET28-Avitag-$Wzc_{JR}$ or pET28-Avitag-$Wzc_{Peri}$ for $Wzc_{JR}$ and $Wzc_{Peri}$ biotinylation in vivo. Cells were grown overnight at 37 °C, and then 1 liter LB broth containing 35 µg/ml Kanamycin and 15 µg/ml chloramphenicol was inoculated with preculture, then grown to an $OD_{600}$ of 0.8 prior to adding 0.2 mM IPTG and 100 µM biotin, and further grown at 18 °C for 16 h. Cells were collected by centrifugation and resuspended in 50 ml of lysis buffer (50 mM Tris pH 7.5, 500 mM NaCl, 10 mM imidazole, 10% glycerol). After adding 1 tablet of protease inhibitor cocktail (11873580001, Roche), cells were lysed by sonication (BANDELIN SONOPLUS; Amplitude 58%; 1 sec on; 8 sec off, 35 min total). After soluble fractions were obtained, batch purification was conducted with 3 ml (bed volume) nickel resin. After 1 h incubation at 4 °C, Nickel resin was washed with 200 ml wash buffer (50 mM Tris pH 7.5, 500 mM NaCl, 20 mM imidazole, 10% glycerol), and bound proteins were eluted in 15 ml (50 mM Tris pH 7.5, 500 mM NaCl, 300 mM imidazole, 10% glycerol). Elution fractions were concentrated using 10 kD cut-off Amicon® Ultra Centrifugal Filters (Merk), and then loaded onto Superdex 200 increase 100/30 GL gel filtration column equilibrated with SEC running buffer (50 mM Tris pH 7.5, 150 mM NaCl, 5% glycerol). The peak elution fractions were first analyzed on SDS-PAGE and then pooled, concentrated, aliquoted, flash frozen, and stored in −80 °C.

The glycan screening assay was performed according to the manufacturer's protocols. The biotinylated recombinant proteins at three different concentrations ($Wzc_{JR}$, 25 µg/ml, 50 µg/ml, and 100 µg/ml; $Wzc_{Peri}$, 50 µg/ml, 100 µg/ml, and 200 µg/ml), were added to array wells, with one buffer-only control well. The samples were incubated overnight with gentle rocking. Glycan-protein binding was detected by incubation with Cy3 equivalent dye-conjugated streptavidin for 1 h at 4 degree. The washing procedure was strictly performed according to the manufacturer's protocols. The glass slide was dried with a nitrogen flow before storage.

The fluorescent signals were scanned using a GenePix 4300 microarray scanner (Molecular Devices) with excitation at 532 nm and emission at 568 nm. Signal intensities were quantified using GenePix Pro 7 software (7.3.0.0). Background correction was performed by subtracting signal values obtained from buffer-only controls to reduce noise. Data were subsequently normalized following the manufacturer's guidelines (GA-Glycan-100-SW). To ensure data quality, four outlier signals (out of 2400 data points) inconsistent with technical replicates were excluded as likely artifacts. Final data visualization, including heatmap generation, was performed using RStudio (version 2024.04.2 + 764).

## Colanic acid production assay
The colanic acid production assay was performed according to previous studies with small modifications[25,50]. To ensure complete extraction of the produced colanic acid, the cell cultures were heated at 95 °C for 30 min, then centrifuged at 20,000 × *g* for 30 min to remove the resulting cell pellets. Briefly, colanic acid was precipitated from the spent M9 culture after boiling and removing the bacterial cells by adding three equivalent volumes of acetone. Added Cys·HCl reacted with fucose to generate a colored product. The concentration of this product was measured colourimetrically to determine the relative concentration of fucose as a proxy for colanic acid concentration.

*E. coli* JM109 (DE3) Δ*wzabc* strain was made from *E. coli* JM109 (DE3) using λred recombination and chloramphenicol resistance cassette as a selection marker[44]. *E. coli* JM109 (DE3) Δ*wzabc* cells were transformed with pET21-Wza-Wzb-Wzc-strep and the derivatives. Pre-

cultures were grown in LB supplemented with appropriate antibiotics at 37 °C overnight. Then the pre-cultures were diluted (1:50) into 1 x M9 media supplemented with 0.4% glucose as a carbon source, and appropriate antibiotics. The cultures were further grown at 19 °C for 1 day to reach around $OD_{600} = 0.6$. Then 0.1 mM IPTG were added to each culture, and cultures were further grown for 24 h. The $OD_{600}$ was measured for all cultures, and the volume of the cultures were normalized based on $OD_{600}$. To ensure complete extraction of the produced colanic acid, the cell cultures were heated at 95 °C for 30 min, and then centrifuged at 20,000 × *g* for 30 min to remove the resulting cell pellets. 4 ×400µl from each sample were taken and transferred into 4 EP tubes, and 100 µl 100% trichloroacetic acid (TCA) was added to each tube to precipitate the protein for 4 h. The protein pellets were discarded after centrifugation at 20,000 x *g* for 30 min, and 450 µl was transferred into a new 2 ml EP tube and 1350 µl acetone was added to precipitate colanic acid overnight. The produced colanic acid was spun down by centrifugation at 20,000 x *g* for 60 min, and air-dried overnight. The purified samples were resuspended in 50 µl Milli-Q $H_2O$. After adding 200 µl of $H_2SO_4/H_2O$ (6:1 v/v), the samples were then heated up to 95 °C for 30 min and then cooled to room temperature. The absorbances of each sample (200 µl) were measured at both 396 and 427 nm within a 96-well U shape plate using a microplate reader (TECAN SPARK Microplate Reader, TECAN). The difference between $OD_{390}$ and $OD_{427}$ is marked as A1. Afterwards, 6 µl freshly prepared cysteine hydrochloride (Cys·HCl, 3% (w/v)) solution was added into each sample well and mixed by pipetting. The absorbances of each sample were measured again at both 396 and 427 nm. The current difference between $OD_{390}$ and $OD_{427}$ is marked as A2. The difference between A2 and A1 was then plotted as a violin plot using graphpad prism 10. Three technical replicates with smaller deviations from the median value were included for the final analysis and bar graph preparation.

## Western-blot analysis
Cell cultures from the colanic acid production assay were normalized according to their $OD_{600}$ values, ensuring equal cell numbers across samples. Equivalent cell aliquots were then mixed with SDS-PAGE loading buffer and incubated at 95 °C for 15 min to denature proteins. Subsequently, 10 µl of each sample was loaded onto an SDS-PAGE gel and electrophoresed at 200 V for 30 min.

Following electrophoresis, proteins were semi-dry transferred onto PVDF membrane at 25 V for 30 min using a Trans-Blot Turbo Transfer System (Bio-Rad). The membrane was then blocked for 30 min in a blocking buffer containing 3% bovine serum albumin (BSA) in 1 × TBST (20 mM Tris-HCl, pH 7.5; 150 mM NaCl; 0.1% [v/v] Tween-20). After blocking, the membrane was washed three times with 1 × TBST and incubated at room temperature for 2 hours with Strep-Tactin AP Conjugate (IBA Lifesciences GmbH) diluted 1:4000 in TBST.

Following incubation, the membrane was washed again three times with 1 × TBST. For colorimetric detection, 5 ml of freshly prepared alkaline phosphatase (AP) buffer (100 mM Tris, pH 8.0; 150 mM NaCl; 1 mM $MgCl_2$) containing 33 µl NBT (Promega) and 66 µl BCIP (Promega) was added to each membrane and incubated until visible color development occurred. The reaction was terminated by rinsing the membranes with distilled water to prevent nonspecific background. Finally, the membrane was scanned for documentation.

## Mass spectrometry
For proteomic analysis, purified wild-type Wzc from SEC was digested using trypsin. For that, 1 µg of protein was solubilized in 100 µl 500 mM TEAB (pH 8) followed by one hour reduction with a final concentration of 5 mM TCEP at 56 °C, and 30 min alkylation with a final concentration of 10 mM MMTS at room temperature. Trypsin (Promega) was then added at a ratio of 0.05 µg Trypsin per 1 µg protein,

and digestion was carried out overnight at 37 °C while shaking at 800 rpm.

Peptides were directly subjected to desalting using the Bravo Automated Liquid Handling Platform (Agilent), applying the standard peptide clean-up method version 3.0 (protein sample prep workbench v.3.2.0) and using reversed-phase S cartridges (G5496-60033, Agilent).

Next, desalted peptides were directly used for phosphopeptide enrichment. Enrichment was conducted on the Bravo system using the standard phosphopeptide enrichment method version 2.1 using Fe(III)-NTA cartridges (G5496-60085, Agilent). In total, eight technical replicates were prepared for LC-MS based phosphoproteome analysis.

For LC-MS analyses peptides were vacuum dried, resuspended in 3 µl of 0.1% formic acid (FA) in water, sonicated for 5 min, and transferred to HPLC vials. LC-MS/MS analyses were performed using a Dionex Ultimate 3000 n-RSLC system (Thermo Fisher Scientific) coupled to an Orbitrap Exploris Hybrid mass spectrometer (Thermo Fisher Scientific). One nanogram of peptides was loaded onto a 50 cm Low-Load µPAC Neo HPLC analytical column (Thermo Fisher Scientific). The peptides were separated at a flow rate of 500 nl/min with a non-linear gradient over 5.5 min from 99% MS buffer A (0.1% FA in water) to 10% MS buffer B (0.1% FA in 80% ACN), followed by 10 min to 35% buffer B and finally 4 min to 98% buffer B at 125 nl/min.

Ionization was achieved via electrospray using a glass emitter (Bruker). Data acquisition was performed using SII software within the Xcalibur suite (v4.4.16.14). The instrument operated in "top speed" mode with a 1.5 s cycle for MS/MS acquisition of doubly and triply charged peptides. Fragmentation was conducted using higher-energy collisional dissociation (HCD), and peptides were measured in the Orbitrap (HCD/OT).

Raw data was analyzed using MaxQuant (v2.6.7.0) against the *Escherichia coli* database (strain K12, downloaded 2025-03-28 from Uniprot). Search parameters included: trypsin as enzyme allowing two missed cleavages; minimum peptide length: 7; maximum modifications on one peptide: 8; fixed modification: β-methylthiolation on cysteine (+ 45.98 Da); variable modifications: oxidation on methionine (+ 15.99 Da) and phosphorylation at serine, threonine, and tyrosine (+ 79.97 Da); search peptide tolerance: 4.5 ppm; false discovery rate (FDR): 1%. Modified and unmodified peptides were used for quantification. Removal of contaminants, missing values and phosphorylated sites with localization probability <85% was done with support of Perseus (v2.1.1.0). Summed intensities of phosphorylation site combinations were calculated based on Peptide Spectrum Match (PSM) representing confirmed identification of peptide sequences. Association analysis (frequent itemset mining approach) was performed in R to evaluate the co-occurrence of phosphorylations (itemsets). Support is defined as the summed intensity of observed modifications normalized to total measured intensities. Lift quantifies deviation from statistical independence, with values > 1 indicating co-enrichment, < 1 suggesting anti-co-occurrence, and 1 reflecting expected random association.

## Protein structure prediction

Protein structures were predicted with a local installation of AlphaFold 3[30]. Protein structure predictions were performed using AlphaFold 3 based on the *google-deepmind/main* repository (commit f407412). To optimize GPU memory usage, the Dockerfile (docker/Dockerfile) was modified to enable unified memory in accordance with the official performance documentation. All predictions were executed using a Singularity container built from the corresponding Docker image. The Singularity image was generated using the command: singularity build alphafold3.sif docker://ntnn19/alphafold3:latest_parallel_a100_80gb.

The protein sequences of putative colanic acid polymerase Wzy (gene name *wcaD*; uniprot ID, P71238) and the co-polymerase Wzc (uniprot ID, P76387) were used for structural predictions.

## Co-sedimentation assay

To evaluate the ability of the sfGFP-Wzc$_{JR}$ fusion protein to bind colanic acid–producing *E. coli*, a co-sedimentation assay was performed using *E. coli* JM109 (DE3) and the isogenic colanic acid-deficient strain JM109 (DE3) Δ*wzabc*. 1 ml of each cell culture from the CPS production assay was harvested by centrifugation (11,000 × g, 1 min). Cell pellets were resuspended in 1× PBS, and purified sfGFP-Wzc$_{JR}$ was added to the cell suspensions at a final concentration of 0.1 mg/ml. Samples were incubated for 10 min at room temperature without shaking to allow binding. Following incubation, cells were collected by centrifugation (11,000 × g, 1 min) and washed twice with 1× PBS to remove unbound protein. The final pellets were resuspended in 1 ml of 1× PBS, and 100 µl of each suspension was transferred into a 96-well microplate. Fluorescence was measured using a TECAN plate reader with an excitation wavelength of 488 nm and emission at 535 nm. Optical density at 600 nm (OD$_{600}$) was measured from the same wells. Fluorescence values were normalized to OD$_{600}$ to account for variations in cell density.

## Statistics & reproducibility

Quantitative experiments were performed in triplicate for deriving statistics. Statistical analyses were performed using GraphPad Prism 10.5.0 (GraphPad, San Diego, CA). Statistical differences between each two groups were determined by using one-way ANOVA by the non-parametric Dunnett test in GraphPad Prism. Statistical differences were defined as $*P \leq 0.05$, $**P \leq 0.01$, $***P \leq 0.001$, and ns $P > 0.05$ (not significant).

## Reporting summary

Further information on research design is available in the Nature Portfolio Reporting Summary linked to this article.

## Data availability

Models and associated cryo-EM maps have been deposited into the PDB database with the following accession codes: EMD-53598, PDB-9R60 (Wza_C1); EMD-53599, PDB-9R61 (Wza_C8); EMD-53600, PDB-9R62 (Wzc$_{K540M}$_C1); EMD-53601, PDB-9R63 (Wzc$_{K540M}$_C4); EMD-53602, PDB-9R64 (Wza-Wzc$_{K540M}$_C1, Conf 0); EMD-53603, PDB-9R65 (Wza-Wzc$_{K540M}$_C8, Conf 0); EMD-53604, PDB-9R66 (Wza-Wzc$_{K540M}$_C1, Conf I); EMD-53605; PDB-9R67 (Wza-Wzc$_{K540M}$_C8, Conf I); EMD-53606, PDB-9R68 (Wza-Wzc$_{K540M}$_C1, Conf II); EMD-53607, PDB-9R69 (Wza-Wzc$_{K540M}$_C8, Conf II); EMD-53608, PDB-9R6A (Wza-Wzc$_{K540M}$_C1, Conf III); EMD-53609; PDB-9R6B (Wza-Wzc$_{K540M}$_C1, Conf IV); EMD-53610; PDB-9R6C (Wza-Wzc$_{K540M}$_C1, Conf V). Previously published PDB codes: 7NII[13]. The mass spectrometry proteomics data have been deposited to the ProteomeXchange Consortium via the PRIDE partner repository[51] with the dataset identifier PXD064872 [http://proteomecentral.proteomexchange.org/cgi/GetDataset?ID=PXD06487]. The source data underlying Figs. 1a–d, 2f, 5d and Supplementary Fig. 1b-f, 9c-f,11a,b are provided as a Source Data file. Source data are provided in this paper.

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

## Acknowledgements

We thank the Heinz group members: Surabhi Lata and Ines Glöckner for support of this project. We thank Sven-Kevin Hotop and Michael Lehky for supporting the scanning glass slides from the glycan array. We also thank Dominik Körner and the Core Facility for Translational Proteomics (CFTP, HZI) for generating and providing the phosphoproteome data and Frank Klawonn for statistical evaluation of phosphosite analyses. Furthermore, we thank Stefan Schmelz for maintaining the cryo-EM facility. We thank Konrad Büssow, Youssef El Mouali Benomar, and Joop Van den Heuvel for kindly providing some plasmids. We thank Natan Nagar for help with large AlphaFold 3 predictions. This project was supported in part through funds available to T.C.M. through the Behörde für Wissenschaft, Forschung und Gleichstellung of the city of Hamburg at the Institute of Microbial and Molecular Sciences at the University Medical Center Hamburg-Eppendorf (UKE) and the Deutsches Elektronen Synchrotron (DESY). C.S. acknowledges support by the Helmholtz Association (VH-NG-1526).

## Author contributions

Conceptualization: B.Y., T.C.M., and D.W.H., cryo-EM sample preparation, data-collection, processing, and model building: B.Y., Structural analysis: B.Y., D.W.H., R.H.J., and T.C.M., Fluorescence microscopy: C.S. and B.Y. Mass spectrometry: T.R., B.Y., and L.J. Glycan array: B.Y. and P.R. Cloning and CPS secretion assay: B.Y. and A.G. Visualization: B.Y., D.W.H., C.S., R.H.J., L.J., and T.C.M. The original draft: B.Y. and D.W.H. Reviewing and editing: all authors.

## Funding

## Competing interests

The author declare no competing interests.
