## [Transparent Peer Review file · Nature Communications]

Molecular insights into the capsular polysaccharide transporter Wza-Wzc complex

Corresponding Author: Professor Dirk Heinz

Version 0:

Reviewer comments:

Reviewer #1

(Remarks to the Author)

What are the noteworthy results and the significance?

The capsular polysaccharide is vital to pathogenesis of many bacteria. There has been a long-standing interest in the assembly, polymerisation and export of the sugar as it has (in gram negative) to cross two membranes. This is of interest both as a target for antibacterials and physical chemistry. The components of the system have been identified, and most have been characterised structurally at an individual level. The frontier has moved to the study of complexes. This paper describes the trans periplasmic Wza (outer membrane) and Wzc (inner membrane) complex. The complex is determined by cryoEM allowing placement of the atoms. The complex is beautiful and its isolation a technical triumph. The complex not only confirms previous hypothesis but crucially it explains in molecularly detail the function. Interestingly the authors identify multiple conformations of the complex. They speculate these conformers are important to function. This This is a very important and significant piece of work.

The authors use the complex to formulate and test some important hypothesis. Wzc contains an insertion relative to its relative Wzz, the paper shows evidence that the insertion (JR domain) is critical in enabling the system. The hypothesis is that JR domain binds the flipped building block carbohydrate and helps transport it towards the polymerase (wzy). This might help rationalise other findings (there has been some speculation / reports suggesting Wzx binds to Wzc to form a large complex; but no direct evidence for protein-protein interaction).

The authors draw out the similarity to the conceptually related Kps system and these results help support our understanding of this system.

Some suggestions to address

It would be good to comment on the likely width of periplasm where the complex forms. There have been some EM images suggesting localisation to poles of these complexes or unusual geometry.

The JR domain hypothesis is interesting, but the evidence is not yet definitive. The challenge is if the sugar binds here how does it get through the solid Wzc ring. There are alternative hypotheses as to the role of the JR domain. If it is important for transporting the substrate to Wzy why is it absent in Wzz?

What influence do the authors think the periplasm has? The assembly of the trans periplasmic complex will be sensitive to the periplasm.

Reviewer #2

(Remarks to the Author)

This manuscript describes the structure of the Wzc-Wza complex that is involved in the synthesis and secretion of capsular polysaccharides by the so-called Wzy-dependent pathway. In this case, short lipid-linked oligosaccharides are shuttled to the periplasmic side of the inner membrane and polymerized into a high molecular weight polymer by the Wzy transferase. Elongation and secretion require the Wzc-Wza complex to mediate CPS transport across the periplasm and the outer membrane.

The manuscript presents various cryo EM structures of the Wzc-Wza macrocomplex, consisting of Wzc and Wza octamers. The expression, purification, and structure determination of this inner and outer membrane-spanning complex are a great technical achievement and provides novel, albeit limited, insights into capsule biogenesis.

Major points:

Figure 1a: The authors use fluorescent imaging of Wza and Wzc to support the claim that (a) Wza forms 'clusters' in the membrane, (b) that Wzc exists in a presumably monomer-oligomer equilibrium, and (c) that Wza colocalizes with Wzc. To make this a solid point, the authors should first clarify what they think 'clusters' refer to (octamers or larger assemblies), perform a more thorough co-localization analysis based on a sufficiently large dataset, and include a suitable negative control (a clustering protein that does not interact with Wzc, for example). Further, it would also be important to analyze the co-localization of Wza/Wzc at endogenous (not over-expressed) expression levels.

Figure 1h - Mutational analysis of Wzc: The panel compares relative CPS levels of specific Wzc mutants but does not compare the expression levels of the individual constructs. Ensuring that the mutants are expressed at similar levels and properly localized is important to determine 'relative' CPS levels. The 'AC' construct carries four point mutations. Do the individual single mutants show any phenotype?

Further, the data shown appear to come from three biological replicates but are merged for graphing and analysis. It would be better to present the data for each replicate and calculate the significance based on the individual averages.

Figure 2 – conformational ensembles: What seems missing from the description of the various Wza-Wzc conformations is a detailed discussion of how the different states differ, specifically in terms of channel properties (structurally and electrostatically) and how these changes could affect CPS polymerization and/or translocation.

Figure 2e and 2f: The conformational changes the authors intend to show in this panel are hard to see without zooming in on the corresponding regions of the complex. This could be improved.

Figure 2i: Please clearly explain in the caption what is shown for Conf III – V, and perhaps revise the coloring to highlight differences between the superimposed protomers.

Figure 3h: It is unclear what the purpose of this panel is. Presumably, it is supposed to show an asymmetric Wzc complex to contrast the C4 symmetric structure determined by the authors. This and the color code should be explained in the caption.

Glycan array study: The glycan-binding experiment is not well described. First, the concentrations of the proteins in the different experiments should be normalized for a direct comparison. Second, the authors do not disclose what glycan structures are recognized by Wzc, and third, it would be informative to know whether there are any common motifs among the glycans recognized.

In addition, because the authors speculate that the JR domain is involved in shuttling CPS oligosaccharides to the interior of the Wzc octamer, can the authors demonstrate that the JR domain indeed binds colanic acid? Since the authors have colanic acid-producing *E. coli*, could the JR domain be labeled and used as a probe?

Figure 5 – Model: The presented model is rather vague and omits critical steps. For example, at what point is the CPS released from the lipid carrier, and how will the relevant enzyme access its substrate inside a Wzc octamer?

Page 8, line 214: The presented AlphaFold model of Wzy inside the Wzc octamer is intriguing. Does Wzy co-purify with Wzc, perhaps after over-expressing both subunits? Additionally, the reported ipTM score of 0.56 indicates a rather low-confidence AlphaFold model. While this does not mean that the model is incorrect, additional experimental validation is certainly necessary, and any supporting literature should be cited.

Figure 2h and Discussion, lines 268-273: There is a reasonable chance that the structure of Wzc associated with two Wza octamers is an artifact due to the overexpression of the components or dissociation and re-association of the binding partners during purification. Therefore, the corresponding discussion should be revised to account for this (likely) scenario.

Minor points:

Line 54: Please explain K30 in terms of its carbohydrate structure.

Line 83: 'Structural heterogeneity and dynamic behavior' is unlikely to be extracted from fluorescent imaging. Please rephrase.

Line 87: Please explain why the 'original ribosome binding site' was maintained in the expression constructs.

Magnesium occupancy of the ADP binding sites: is it possible that the Mg²⁺ ion is not resolved in some states due to heterogeneity? This could be tested by symmetry expansion and careful sorting of the protomers.

Line 141-142: It is proposed that the Wzc conformational transitions may generate a mechanical force that could translocate the CPS. First, this idea could be moved to the Discussion and second, it would require some supporting information. For example, are there any sites inside the tunnel that could interact with the CPS, and if so, are these sites affected by the structural reorientations?

Can the authors compare the electrostatic properties of the channel interior in all observed states? Also, a careful comparison of conformation 0 and IV could reveal critical transitions associated with Wza binding.

Line 197: 'Ion leakage' Where would the ions come from, and which membrane would they leak across? The outer membrane should be permeable to most small solutes and ions. Is there evidence that Wzc conducts ions across the inner membrane?

Discussion, line 316: The authors state that Wzc exists in a monomer-oligomer equilibrium when phosphorylated. However, the presented data do not address this point. If this statement is based on previously published data, this should be clarified. To address this point, could the Y-tail be removed or replaced, followed by fluorescent imaging?

Discussion, last paragraph, lines 313 - 328: Please rephrase to emphasize that this paragraph presents a hypothetical model.

Methods: Description of cryo EM data processing, lines 541-543: This section is fairly short, which is ok if the data processing was straightforward. If not, technical difficulties related to particle picking, sorting, or focused refinements should be described.

Methods, glycan array, line 545 – 547: Please state the vendor and catalogue number of the arrays used.

Version 1:

Reviewer comments:

Reviewer #2

(Remarks to the Author)

The revised manuscript by Biao Yuan et al. has improved significantly. The authors have addressed most of our concerns, although some remain.

1. The co-localization analysis of Wza and Wzc is still not ideal. Including the D1 mutant was a good idea, but the concern regarding the overexpression of the components remains. To this end, the authors titrated IPTG to affect the expression levels of the binding partners. How do we know that the expression levels are indeed reduced at 10 versus 40 uM IPTG? The authors state that 10 uM corresponds to approx. endogenous expression levels, but how do we know?
2. Mutational analysis of Wzc: We recommended to include a comparison of the expression levels of the individual Wzc mutants. To this end, the authors included a Western blot as a supplemental figure (1d). However, it is not stated what antibody was used and, more importantly, how the samples were normalized. Was this based on optical density? Ideally, one would include a blot for a housekeeping protein as a loading control.
3. JR domain binding to colanic acid. Based on our suggestion, the authors tested whether the JR domain binds to colanic acid on the cell surface but failed to detect binding. We appreciate that this was attempted. However, this information is not included or discussed as part of the model. It seems that microarray data suggest binding to certain motifs, yet the observation that the JR domain does not bind to the assembled CPS may tell us something about what it really recognizes. Therefore, we suggest including this negative data in the overall description of carbohydrate binding to the JR domain.
4. Prompted by our skepticism, the authors now state that the dimeric Wza complex associated with one Wzc is unlikely to be an in vitro artefact. Apparently, one supporting argument is that Wza tends to form 2D lattices. However, the arrangement of Wza in conformation V is inconsistent with a planar 2D lattice because the copies are not coplanar. Thus, we still caution overinterpreting the biological significance of conformation V.

Version 2:

Reviewer comments:

Reviewer #2

(Remarks to the Author)

The authors have addressed our concerns to the best of their abilities. We have no further comments.

made.

REVIEWER COMMENTS

Reviewer #1 (Remarks to the Author):

What are the noteworthy results and the significance?

The capsular polysaccharide is vital to pathogenesis of many bacteria. There has been a long-standing interest in the assembly, polymerisation and export of the sugar as it has (in gram negative) to cross two membranes. This is of interest both as a target for antibacterials and physical chemistry. The components of the system have been identified, and most have been characterised structurally at an individual level. The frontier has moved to the study of complexes. This paper describes the trans periplasmic Wza (outer membrane) and Wzc (inner membrane) complex. The complex is determined by cryoEM allowing placement of the atoms. The complex is beautiful and its isolation a technical triumph. The complex not only confirms previous hypothesis but crucially it explains in molecularly detail the function. Interestingly the authors identify multiple conformations of the complex. They speculate these conformers are important to function. This is a very important and significant piece of work.

The authors use the complex to formulate and test some important hypothesis. Wzc contains an insertion relative to its relative Wzz, the paper shows evidence that the insertion (JR domain) is critical in enabling the system. The hypothesis is that JR domain binds the flipped building block carbohydrate and helps transport it towards the polymerase (wzy). This might help rationalise other findings (there has been some speculation / reports suggesting Wzx binds to Wzc to form a large complex; but no direct evidence for protein-protein interaction).

The authors draw out the similarity to the conceptually related Kps system and these results help support our understanding of this system.

We sincerely thank the reviewer for the enthusiastic assessment of our work. We are delighted that the reviewer recognizes the significance of the successful isolation and structural characterization of entire CPS secretion machinery. We are grateful for the constructive comments, which further motivate us to refine the manuscript for clarity and impact and to continue exploring this molecular machine in the future.

Some suggestions to address:

It would be good to comment on the likely width of periplasm where the complex forms. There have been some EM images suggesting localisation to poles of these complexes or unusual geometry.

We thank the reviewer for this important point. Cryo-ET measurements place the average periplasmic width of *E. coli* at ~332 Å (± 17.6 Å) (DOI: 10.1128/spectrum.01290-22). Notably, the periplasm is not uniform, with local variations of ~210-440 Å observed by cryo-ET (DOI: 10.1128/spectrum.01290-22). Besides, the ~250 Å periplasmic spanning of the Wza-Wzc complex is consistent with other periplasm-spanning assemblies such as the type III secretion injectisome (~250 Å; DOI: 10.1038/s41467-021-21143-1), the type IV secretion system (~210 Å; DOI: 10.1038/s41586-022-04859-y) and the AcrAB–TolC efflux pump (~210 Å; DOI: 10.1038/nature13205). These complexes are known to induce local narrowing of the periplasm by bending or compressing the inner and outer membranes around the conduit. Such constriction is thought to facilitate efficient translocation by minimizing diffusion losses and forming a continuous, protected passage across the envelope.

Colocalization was observed between clustered forms of Wza and Wzc with no noticeable spatial bias (Fig. 1a), indicating that CPS transporters do not exhibit a preferred cellular localization, similarly to the AcrAB–TolC complex which is uniformly distributed across the cell envelope, and in contrast to the type IV secretion system in *Legionella pneumophila*, which often exhibits polar enrichment (DOI: 10.1038/s41564-018-0165-z).

We have integrated this information into the Results and Discussion section.

Results section, p. 5: “Notably, colocalization was observed between clustered forms of Wza and Wzc with no noticeable spatial bias (**Fig. 1a**), suggesting that these CPS transporters are uniformly distributed in the cell envelope.”

Discussion section, p. 14: “The periplasmic width of *E. coli* is approximately 330 Å, though this spacing is not uniform and can vary between ~210-440 Å depending on envelope stress and the presence of trans-envelope assemblies¹⁷. Within this physiological range, the ~250 Å span of the Wza-Wzc complex is structurally plausible and comparable to other envelope-spanning machines, including the type III secretion system injectisome (~250 Å)³⁵, type IV secretion system (~210 Å)³⁶ and the AcrAB–TolC efflux pump (~210 Å)³⁷. Such complexes are thought to locally constrict the periplasm by bending and compressing the membranes, thereby establishing a continuous conduit across the envelope. This local narrowing may enhance transport efficiency by reducing diffusion losses and providing a protected passage for substrates.”

The JR domain hypothesis is interesting, but the evidence is not yet definitive. The challenge is if the sugar binds here how does it get through the solid Wzc ring. There are alternative hypotheses as to the role of the JR domain. If it is important for transporting the substrate to Wzy why is it absent in Wzz?

We concur with the reviewer that our proposed model for the JR domain remains hypothetical. Two primary mechanisms have been proposed for polysaccharide co-polymerase (PCP) function: (1) PCP-1 Wzz acting as a scaffold that mediates chain transfer among multiple Wzy molecules, with O-polysaccharide (OPS) elongation proceeding externally until the growing

chain disrupts Wzz interaction (DOI: 10.1016/j.str.2017.03.017); and (2) O-antigen polymerization occurring within the Wzz lumen (DOI: 10.1098/rsob.220373). PCP-2a Wzc differs from PCP-1 Wzc by possessing both a cytoplasmic kinase domain and the JR domain. Although deletion of the JR domain does not completely abolish CPS production, it markedly decreases the yield. Structural superimposition of Wzc_{K540M} with the mannohexaose-bound CBM29-2 structure (PDB: 1GWL) revealed close alignment of the JR domains, with the bound mannohexaose suggesting a potential entry site for CPS repeat units. Together with its CBM-like features, these observations support the idea that the JR domain, while not essential, enhances CPS biosynthesis efficiency by facilitating the loading of CPS repeat units into the polymerization machinery.

We have put this information into the Discussion section.

Discussion section, p. 16-17: “Two main mechanisms have been proposed for the function of polysaccharide co-polymerase 1 (PCP 1) Wzz, either serving as a scaffold that coordinates chain elongation among multiple Wzy molecules or acting as a lumen within which polymerization occurs^{10,41}. Unlike PCP-1 Wzc, PCP-2a Wzc contains an additional cytoplasmic kinase and JR domain (Supplementary Fig. 10)^{13,29}. Although JR deletion does not abolish CPS production, it substantially reduces yield, indicating a supportive rather than essential role. Thus, we propose JR domain functions as a carbohydrate-recognition module that facilitates the capture and delivery of CPS repeat units to the polymerization platform. The proposed substrate entry site, formed by adjacent Wzc protomers, provides a mechanistic explanation for how CPS units are funneled into the inner membrane cavity for elongation”

What influence do the authors think the periplasm has? The assembly of the trans periplasmic complex will be sensitive to the periplasm.

The periplasmic environment likely affects the assembly and stability of trans-envelope complexes through factors such as width, peptidoglycan constraints, ionic strength, and redox state (DOI: 10.1371/journal.pbio.2004935; DOI: 10.1016/j.bpj.2021.08.016; DOI: doi.org/10.3389/fmolb.2020.00166; DOI: 10.1074/jbc.M110.119321). Conversely, the formation of large trans-envelope machines such as the T3SS, T4SS, or AcrAB–TolC can actively remodel the periplasm by inducing curvature in the inner and outer membranes, thereby narrowing the intervening space (DOI: 10.1038/ncomms10114; 10.1038/s41467-019-10512-6). Such localized constriction may minimize diffusion losses, shield translocating substrates, and influence the distribution of periplasmic crowding agents and ions. Thus, the relationship between the complex and its compartment is bidirectional: the periplasm dictates assembly dynamics, while the assembled complex, in turn, reshapes the periplasmic landscape to promote efficient transport. In our system, because the Wza–Wzc complex is primarily stabilized by electrostatic interactions (Fig. 2), changes in the periplasmic ionic strength are likely to play a particularly important role in modulating its assembly. Although these parameters were not directly measured in our study, they represent an important avenue for future investigation.

We have put this information into the Discussion section.

Discussion section, p. 13: “In addition, the Wza-Wzc complex is primarily stabilized by electrostatic interactions, suggesting that fluctuations in periplasmic ionic strength could significantly impact its assembly dynamics. Although these parameters were not directly assessed here, they represent an important factor that may fine-tune the efficiency and regulation of CPS export.”

Reviewer #2 (Remarks to the Author):

This manuscript describes the structure of the Wzc-Wza complex that is involved in the synthesis and secretion of capsular polysaccharides by the so-called Wzy-dependent pathway. In this case, short lipid-linked oligosaccharides are shuttled to the periplasmic side of the inner membrane and polymerized into a high molecular weight polymer by the Wzy transferase. Elongation and secretion require the Wzc-Wza complex to mediate CPS transport across the periplasm and the outer membrane.

The manuscript presents various cryo EM structures of the Wzc-Wza macrocomplex, consisting of Wzc and Wza octamers. The expression, purification, and structure determination of this inner and outer membrane-spanning complex are a great technical achievement and provides novel, albeit limited, insights into capsule biogenesis.

We thank the reviewer for acknowledging the technical challenges associated with expressing, purifying, and resolving the structure of the Wzc–Wza machinery. To our knowledge, our cryo-EM structures of the intact CPS transporter provide the first near-atomic visualization of this trans-envelope assembly, offering a structural framework for understanding CPS biogenesis. While we agree that our current insights into the CPS polymerization platform are limited, we believe they lay the foundation for future mechanistic and functional investigations.

Major points:

Figure 1a: The authors use fluorescent imaging of Wza and Wzc to support the claim that (a) Wza forms ‘clusters’ in the membrane, (b) that Wzc exists in a presumably monomer-oligomer equilibrium, and (c) that Wza colocalizes with Wzc. To make this a solid point, the authors should first clarify what they think ‘clusters’ refer to (octamers or larger assemblies), perform a more thorough co-localization analysis based on a sufficiently large dataset, and include a suitable negative control (a clustering protein that does not interact with Wzc, for example). Further, it would also be important to analyze the co-localization of Wza/Wzc at endogenous (not over-expressed) expression levels.

We now expanded our dataset to include a larger number of cells and imaging fields to ensure robust statistical power. In addition, we will include an appropriate negative control.

Wza is a well-characterized outer membrane protein that functions as a remarkably stable octameric complex (DOI: 10.1038/nature05267). Previous studies have demonstrated that *E. coli* K30 Wza octamer is highly resistant to SDS treatment (DOI: 10.1093/emboj/19.1.57) and has a strong tendency to form 2D crystalline (DOI:10.1074/jbc.M308775200). We observed that *E. coli* K12 Wza also exhibits similar properties in solution based on the cryo-EM analysis (Supplementary Fig. 6a), indicating that Wza may assemble these octamers into larger supramolecular structures.

Consistent with these observations, our fluorescence microscopy analysis of Wza clusters shows a range of fluorescence intensities. The predominant population exhibits relative intensities around 1, likely corresponding to individual octameric complexes. In addition, a smaller but distinct subset of clusters displays higher intensities (ranging from 2 to 5), suggesting higher-order assemblies. This pattern aligns well with the known crystallization tendency of Wza, supporting the interpretation that the observed clusters correspond to both stable octamers and supramolecular arrangements, rather than nonspecific aggregates.

As a negative control, we employed a D1 domain deletion construct (Wza_{ΔD1}), since the D1 domain of Wza has previously been predicted to engage in complex formation with Wzc (DOI: 10.1038/s41467-021-24652-1). This is clearly confirmed by our cryo-EM structure of the Wza-Wzc complex. The deletion mutant shows a very weak Pearson's correlation coefficient (PCC score, 0.1) score, indicating a complete loss of the specific Wza-Wzc interaction. While the weak residual overlap in fluorescence signals likely reflects the limited spatial resolution of fluorescence microscopy, which can cause apparent overlap of Wza and Wzc signal intensities rather than true colocalization. In addition, the presence of monomeric Wzc distributed across the envelope may contribute to background signal and further apparent noise in the analysis.

To exclude potential artifacts arising from overexpression, we also examined colocalization at lower IPTG concentrations (10 μM, 20 μM, and 40 μM), approximating near-endogenous expression levels. Similar PCC scores were observed under all test conditions, confirming that the detected colocalization is not an artifact of overexpression.

The data were shown in Fig. 1 and Supplementary Fig. 6a. The corresponding text has been added as follows:

Results section, p. 5-6: “Wza forms a well-characterized, stable octameric complex that functions as an outer membrane CPS translocon¹⁹. Previous studies have shown that the Wza octamer is highly resistant to SDS treatment²³ and exhibits a pronounced tendency to form two-dimensional crystalline arrays²². These properties indicate that Wza can exist not only as discrete octameric units but also as higher-order assemblies. In line with these findings, our fluorescence microscopy analysis of Wza clusters reveals a broad distribution of fluorescence intensities (**Fig. 1d**). The predominant population exhibits relative intensities around 1, most likely representing single octameric complexes. A smaller yet distinct subset displays higher intensities (ranging from 2 to 5), suggesting higher-order assemblies. This finding aligns well with the known crystallization propensity of Wza and supports the interpretation that the observed clusters represent both stable octamers and supramolecular arrangements.”

Results section, p. 10: “In addition, some micrographs of *E. coli* K12 Wza revealed 2D crystalline arrangements (**Supplementary Fig. 6a**), consistent with those observed for *E. coli* K30 Wza²².”

Results section, p. 5: “The D1 domain of Wza has previously been predicted to mediate interactions with Wzc¹³. Therefore, we generated a deletion mutant Wza_{ΔD1} as a negative control. Although Wza_{ΔD1} was still assembled into well-defined clusters, the markedly low Pearson’s correlation coefficient (PCC = 0.1) confirmed a complete disruption of the Wza-Wzc interaction. (**Fig.1b and c**). To exclude potential artifacts from protein overexpression, we assessed colocalization at reduced IPTG concentrations (10 μM, 20 μM). A similar trend was observed under both conditions, demonstrating that the colocalization is not attributable to overexpression (**Fig. 1c**).”

Methods section, p. 25-26: “protein expression was induced with varying concentrations of IPTG (10 μM, 20 μM, and 40 μM) and the cultures were incubated overnight at 19 °C.Widefield fluorescence imaging for cluster quantification was performed using a Nikon Ti Eclipse N-STORM microscope equipped with laser lines at 488 nm (Sapphire, Coherent) and 561 nm (Sapphire, Coherent). The sample was observed through a Nikon Apo TIRF 100x Oil DIC N2 NA 1.49 objective, and emitted light was detected on an Andor iXon3 DU-897 EMCCD camera (Oxford Instruments).”

Colocalization analysis and protein cluster quantification

Multicolor fluorescence microscopy images taken on the LSM980 Airyscan2 were processed using Fiji⁴⁵. Colocalization was quantified using the Fiji plugin EzColocalization⁴⁶. The whole field of view was analyzed following a manual intensity segmentation to specifically focus on the clustered fraction of Wza-mScarlet3. Diffraction-limited clusters of Wza-mScarlet3 were imaged using a Nikon Ti Eclipse N-STORM microscope. The clusters were identified and quantified using the ThunderSTORM⁴⁷ localization plugin for Fiji. ”

Figure 1h - Mutational analysis of Wzc: The panel compares relative CPS levels of specific Wzc mutants but does not compare the expression levels of the individual constructs. Ensuring that the mutants are expressed at similar levels and properly localized is important to determine ‘relative’ CPS levels. The ‘AC’ construct carries four point mutations. Do the individual single mutants show any phenotype?

To address these important points, we generated the corresponding single mutant constructs, and carefully revised our assay following the methodology described by Obadia et al. (DOI: 10.1016/j.jmb.2006.12.048), which presented a more rigorous quantification of colanic acid production. Specifically, to obtain a more accurate and reliable measurement, we quantified total colanic acid rather than limiting our analysis to the fraction secreted into the medium, as a significant portion can remain loosely associated with the cell surface.

We performed western blot analyses to compare the expression levels of WT Wzc and its variants, and systematically quantified their effects on colanic acid production. As shown in **Supplementary Fig. 1d**, all mutant proteins are expressed at levels comparable to WT Wzc. This refined approach provides a more robust assessment of the phenotypic consequences of the mutations. Importantly, the new data demonstrate that substitution of charged residues at the AC interface nearly abolished colanic acid production, strongly reinforcing the conclusion that electrostatic interactions are critical for stabilizing the Wza-Wzc complex. We are grateful for the reviewer’s insightful comments, which motivated these additional experiments and methodological refinements, thereby significantly strengthening the mechanistic basis of our study.

d

The new data were incorporated in Supplementary Fig. 1d, and Fig. 2f. We also added text as follows:

Results section, p. 7: “To assess the functional impact of the mutations on CPS production, we performed a standard colanic acid extraction and quantification assay²⁵ using an *E. coli* JM109 (DE3) $\Delta wzabc$ strain complemented with plasmids expressing the *wzabc* operon carrying the respective Wzc variants. All tested Wzc variants exhibited expression levels similar to those observed for the wild-type protein (**Supplementary Fig. 1d**). As shown in **Fig. 2f**, quantitative analysis revealed differential levels of CPS production among the Wzc variants. Notably, the AC interface mutations ($WzC_{K332E,Y334A,T335A,H338E}$) completely abolished CPS production, with the charged residue pairs D99-K322 and H338-E102 contributing to the electrostatic interactions, underscoring the essential role of the Wza-Wzc complex in the CPS production. In contrast, substitutions at Y334 and T335 had little to no effect, with the latter even enhancing CPS production.”

Methods section, p. 30: “To ensure complete extraction of the produced colanic acid, the cell cultures were heated at 95 °C for 30 min, and then centrifuged at 20,000 × g for 30 min to remove the resulting cell pellets.”

Further, the data shown appear to come from three biological replicates but are merged for graphing and analysis. It would be better to present the data for each replicate and calculate the significance based on the individual averages.

In the revised figures, we now present the mean of the three technical replicates as a single data point for each biological replicate. Thus, three biological replicates are shown for each condition. Statistical significance was calculated based on these biological replicate means using a one-way ANOVA in GraphPad Prism.

The changes were incorporated into Figure legend for Fig.2f .

Figure legends section, p. 35: “CPS (colanic acid) production levels are presented as filled black circles, representing the mean \pm s.d. from three independent biological replicates. Each circle denotes the average of three technical replicates. Statistical analysis was performed on biological replicate means using one-way ANOVA in GraphPad Prism.”

Figure 2 – conformational ensembles: What seems missing from the description of the various Wza-Wzc conformations is a detailed discussion of how the different states differ, specifically in terms of channel properties (structurally and electrostatically) and how these changes could affect CPS polymerization and/or translocation.

To follow this important suggestion, we have conducted a more rigorous structural comparison of the different Wza-Wzc conformational states (Confs I–IV) followed by a discussion on their potential implications for CPS polymerization and translocation.

The new analysis data were incorporated into Supplementary Fig. 5c-f. The following text was added into the Results and Discussion sections.

Results section, p. 8: “Structural superposition of the Wza regions from Conf I and Conf II revealed that the Wza translocon remains largely conformationally stable (**Supplementary Fig. 5a**). Alignment of the structures on Wza showed that Wzc rotates by about 25° and each protomer undergoes an overall displacement of ~20 Å. This coordinated motion suggests that the Wzc HA domain drives the extension of the CPS secretion machinery through a twisting mechanism (**Fig. 3e and Supplementary Fig. 5b**).

.....Electrostatic mapping of the Wzc periplasmic domain revealed a large hydrophilic channel with minor changes in charge distribution among Confs I – IV, whereas the channel geometry varied substantially (**Supplementary Fig. 5c-f**). The balanced distribution of positively (e.g., K307, K326) and negatively charged residues (e.g., D301, E329, E380) within the channel likely contributes to minimizing non-specific electrostatic interactions with the negatively charged CPS polymer. Comparison of Conf I (channel width of ~17 Å at K307; ~36 Å at E329) with Conf II (~13 Å and ~31 Å) indicates channel narrowing and charge rearrangement at the AC interface, leading to compaction of the forearm region and relaxation of the upper arm (**Supplementary Fig. 5c-d**). Conf III corresponds to a blocked channel conformation (**Supplementary Fig. 5e**), whereas Conf IV displays partial assembly via four forearms, likely representing a transitional intermediate (**Supplementary Fig. 5f**).

Collectively, these results demonstrate the intrinsic flexibility of the Wzc HA domains, particularly in the forearm regions (**Fig. 3i**), and these structural transitions support a twisting-driven model in which Wzc rotation dynamically modulates channel architecture to guide CPS export.”

Discussion section, p. 15: “Multiple conformational changes via the periplasmic domains of Wzc provide molecular insights into how the Wza-Wzc machinery mediates CPS secretion through coordinated structural changes. Wza-Wzc forms a broad aqueous channel that extends across the periplasm, enabling the translocation of a branched, flexible, high molecular weight CPS polymer³⁸. The twisting-driven motion of Wzc and the accompanying modulation of the periplasmic channel from Conf I to Conf II reveal a coordinated mechanism in which mechanical rotation, and pore-size adjustments act together to facilitate CPS translocation. Importantly, this elongation from Conf I to Conf II likely generates a mechanical force sufficient to extract the final CPS repeat unit from the inner membrane, thereby aiding its dissociation from Wzy. By coupling torsional motion to channel constriction, Wzc may act as a molecular winch that pulls out CPS polymer from the polymerization platform. Conf III likely corresponds to the loading state of CPS from Wzc to Wza, where the forearms occupy the channel lumen, closing the inner membrane cavity and halting additional CPS synthesis. The side-open channel of Conf IV exposes the inner membrane cavity, probably accommodating the growing CPS polymer. Once the polymer reaches full length, the channel completes its assembly to facilitate CPS export.”

Figure 2e and 2f: The conformational changes the authors intend to show in this panel are hard to see without zooming in on the corresponding regions of the complex. This could be improved.

Figures 2e and 2f have been revised accordingly to provide enlarged views of the relevant regions of the complex. These changes make the conformational differences more apparent and improve the overall clarity of the figures.

Now the figures were presented in Fig.3e and f, and Supplementary Fig. 5a and b:

Figure 2i: Please clearly explain in the caption what is shown for Conf III – V, and perhaps revise the coloring to highlight differences between the superimposed protomers.

We have updated the caption of Figure 2i to clearly describe conformations III–V, while retaining the original color scheme of the Wzc protomers to ensure consistency with panels a–h.

Figure legends section, p37: **(i) Comparison of Wzc protomers across Confs I – V.** Conf I shows one protomer from the Conf I state; Conf II shows one protomer from the Conf II state; Conf III shows a superposition of all 8 protomers from the Conf III state; Conf IV shows a superposition of all 8 protomers from the Conf IV state; Conf V shows a superposition of all 8

protomers from the Conf V state. Wzc protomers are colored according to the same scheme used in panels a–h.”

Figure 3h: It is unclear what the purpose of this panel is. Presumably, it is supposed to show an asymmetric Wzc complex to contrast the C4 symmetric structure determined by the authors. This and the color code should be explained in the caption.

Figure 3h intended to show an asymmetric *E. coli* K30 Wzc complex in contrast to the C4-symmetric *E. coli* K12 Wzc model obtained by C1 reconstruction.

The caption has been updated to clarify this.

Figure legends section, p. 39: “(f-g) Structural model of *E. coli* K12 Wzc_{K540M} octamer generated based on the C1-reconstructed EM density map.”

Glycan array study: The glycan-binding experiment is not well described. First, the concentrations of the proteins in the different experiments should be normalized for a direct comparison. Second, the authors do not disclose what glycan structures are recognized by Wzc, and third, it would be informative to know whether there are any common motifs among the glycans recognized.

We evaluated each protein across a range of concentrations to examine the consistency of glycan binding. Both proteins were tested within comparable molar ranges: Wzc_{JR} (~15 kDa) at 25–100 µg/mL (≈1.67–6.67 µM) and Wzc_{peri} (~45 kDa) at 50–200 µg/mL (≈1.11–4.44 µM). For this reason, normalization was not applied. Within these concentration ranges, the overall signal intensities and ranked glycan binding preferences remained stable, indicating that specificity was not influenced by protein concentration. Glycan array profiling revealed 13 glycans (IDs 2, 3, 5, 8, 9, 28, 39, 52, 53, 56, 85, 97, 99) that consistently bound both Wzc_{peri} and Wzc_{JR} proteins across replicates. The structures of these glycans, corresponding to the manufacturer’s Glycan Array 100 IDs, are now included in Supplementary Table S4. The following text was added to the Results section.

Results section, p.13: “Glycan array profiling revealed 13 glycans (**Supplementary Table 4**) consistently bound to both the Wzc_{peri} and Wzc_{JR} (**Supplementary Fig. 9**). These glycans are notably enriched in N-acetylglucosamine (GlcNAc), galactose (Gal), and mannose (Man) residues, indicating a recognition preference centered around GlcNAc and Gal. The recurring presence of these sugars suggests that the GlcNAc-Gal linkage serves as a key recognition scaffold. Structurally, the bound glycans closely resemble the repeating-unit composition of colanic acid, which includes Gal-GlcNAc-Fuc-GlcA. Furthermore, consistent binding to aminoglycosides and acidic sugars points to an electrostatic contribution to glycan recognition.

Altogether, these results suggest a GlcNAc-Gal-based carbohydrate recognition motif reinforced by charge-mediated interactions, supporting that Wzc JR domain contributes to CPS assembly.”

In addition, because the authors speculate that the JR domain is involved in shuttling CPS oligosaccharides to the interior of the Wzc octamer, can the authors demonstrate that the JR domain indeed binds colanic acid? Since the authors have colanic acid-producing *E. coli*, could the JR domain be labeled and used as a probe?

To determine whether the JR domain can directly interact with colanic acid, we constructed and purified a GFP-Wzc_{JR} fusion protein. We then assessed its ability to co-sediment with *E. coli* JM109 (DE3) cells that produce colanic acid and with *E. coli* JM109 (DE3) Δ wzabc cells, which lack colanic acid production. However, TECAN-based fluorescence measurements showed no significant difference in fluorescence intensity between the two strains following co-sedimentation. This result may indicate that the JR domain binds colanic acid only transiently, and any loosely associated colanic acid was likely removed during the PBS washing steps. We appreciate this insightful suggestion and plan to further investigate the potential interaction between the JR domain and colanic acid using complementary approaches, such as fluorescence microscopy.

Figure 5 – Model: The presented model is rather vague and omits critical steps. For example, at what point is the CPS released from the lipid carrier, and how will the relevant enzyme access its substrate inside a Wzc octamer?

We have revised and refined the model to include the missing mechanistic details. The updated version is now presented in **Fig. 6**, where we illustrate how Wzy gains access to its substrate within the Wzc octameric complex, and clarify the timing of CPS release from the lipid carrier.

The corresponding Figure legend and Discussion section were also updated as follows:

Figure legends, p. 43: **“Fig. 6. Proposed model of CPS polymerization and secretion. (a)** Preassembly state of the CPS translocation machinery: The Wza translocon is preassembled as an octamer in the outer membrane (OM). Wzc is located in the inner membrane (IM) as an autophosphorylated monomer, with its HA domain adopting a conformation that prevents interaction with Wza. The cytoplasmic phosphatase Wzb can access and dephosphorylate Wzc. **(b)** Initiation of CPS polymerization platform assembly: Dephosphorylation by Wzb triggers Wzc oligomerization, likely around the CPS polymerase Wzy, resulting in the formation of a functional polymerization platform. **(c)** Recognition and loading of CPS repeat units: CPS repeat units, translocated from the cytoplasmic to the periplasmic side by Wzx, are recognized by the JR domain of Wzc and subsequently delivered to Wzy within the inner-membrane cavity of Wzc for polymerization. **(d)** CPS polymerization: As the CPS polymer elongates along the HA domain, the HA domain adopts an extended “forearm conformation”, allowing recruitment of the preassembled Wza translocon. **(e)** Assembly of the CPS translocation machinery: Upon CPS polymer maturation the HA domains of the Wzc octamer open to form a continuous secretion channel together with the Wza translocon, spanning the entire periplasm. **(f)** CPS release: Twisting of the HA domains of Wzc extends the secretion channel and may generate the mechanical force required to drive CPS release from the inner membrane into the Wza translocon. **(g)** Disassembly of the CPS secretion complex: The mature CPS is exported across

the outer membrane *via* Wza, after which the secretion machinery disassembles through Wzc auto-phosphorylation.”

Discussion section, p. 18: “As polymer elongation advances, conformational rearrangements within the HA domain of Wzc promote exposure of its forearm conformation, enabling the formation of a continuous secretion conduit that connects the polymerization site to the extracellular space. The gradual alignment of the Wza-Wzc complex likely ensures synchronized polymer elongation and export, preventing periplasmic accumulation of intermediate-length CPS chains. Twisting of the HA domain induces an approximately 25° rotation of the Wzc octamer within the inner membrane. This coordinated motion most likely exerts mechanical tension along the HA domain, promoting the release of the CPS polymer from Wzy.”

Page 8, line 214: The presented AlphaFold model of Wzy inside the Wzc octamer is intriguing. Does Wzy co-purify with Wzc, perhaps after over-expressing both subunits? Additionally, the reported ipTM score of 0.56 indicates a rather low-confidence AlphaFold model. While this does not mean that the model is incorrect, additional experimental validation is certainly necessary, and any supporting literature should be cited.

In our study, we encountered challenges in expressing Wzy (designated as WcaD in our system), which prevented experimental validation of potential co-purification with Wzc. We acknowledge that the reported ipTM score of 0.56 reflects only moderate confidence in the predicted interface. Consequently, we have emphasized that the proposed Wzy-Wzc model should be regarded as a preliminary structural hypothesis. This clarification, along with the citation to the relevant study (DOI: 10.1098/rsob.220373) supporting the potential Wzy-Wzc interaction, has been incorporated into the revised manuscript.

Results section, p. 11: “Our AlphaFold prediction suggests with moderate confidence (ipTM score: 0.56) that CPS polymerization may take place on an inner membrane Wzy-Wzc platform resembling the previously described WzzE–WzyE complex¹⁰.”

Figure 2h and Discussion, lines 268-273: There is a reasonable chance that the structure of Wzc associated with two Wza octamers is an artifact due to the overexpression of the components or dissociation and re-association of the binding partners during purification. Therefore, the corresponding discussion should be revised to account for this (likely) scenario.

We agree that complex rearrangements involving dissociation and re-association following EDTA treatment are plausible, as we detected a markedly reduced proportion of particles containing a single Wzc molecule associated with two Wza translocons. Additionally, Wza tends to form two-dimensional crystalline arrays *in vitro*, and fluorescence microscopy revealed higher-order Wza clusters *in vivo* even under very low induction levels (40 μM IPTG). These observations suggest that the detected assembly is unlikely to be an experimental artifact, but instead may represent a physiologically relevant or transient intermediate state of the Wza–Wzc complex. The Discussion has been revised to clarify this interpretation and to emphasize that

further studies employing higher-resolution imaging approaches, such as cryo-ET and MINFLUX nanoscopy, will be necessary.

Discussion section, p. 15: “An intriguing finding from our study is the identification of a Wzc octamer associated with two Wza translocons (Conf V), suggesting potential rearrangements through dissociation and re-association or the presence of alternative assembly states. Considering the established propensity of Wza to form two-dimensional crystalline arrays *in vitro* (**Supplementary Fig. 6a**)²² and higher-order clusters *in vivo* under low induction conditions (**Fig. 1d**), this dual-translocon arrangement is unlikely to be an experimental artifact. Instead, it may represent a physiologically relevant or transient intermediate that enables dynamic modulation of HA domain interactions between translocons, thereby promoting efficient CPS secretion. Further *in vivo* studies and high-resolution imaging techniques, such as cryo-ET and MINFLUX nanoscopy, will be essential to determine whether such configurations occur naturally and contribute to CPS export in bacterial systems.”

Minor points:

Line 54: Please explain K30 in terms of its carbohydrate structure.

The text describing the K30 capsular polysaccharide (CPS) has now been included in the revised manuscript.

Introduction section, p. 4: “The coupled secretion machinery is encoded by the *wzabc* operon in *E. coli* K30 strain, which produces the K30 CPS composed of repeating units featuring a $\rightarrow 2$ - α -D-Manp-(1 \rightarrow 3)- β -D-Galp-(1 \rightarrow backbone, with a β -D-GlcA-(1 \rightarrow 3)- α -D-Galp-(1 \rightarrow disaccharide branch linked to the 3-position of the backbone mannose^{7,18}.”

Line 83: ‘Structural heterogeneity and dynamic behavior’ is unlikely to be extracted from fluorescent imaging. Please rephrase.

We rephrased the text accordingly.

Results section, p. 5: “By contrast, Wzc exhibited two distinct distribution patterns: localized clusters and an evenly distributed membrane-associated signal, suggesting the presence of multiple organizational states. Notably, colocalization was only observed between clustered forms of Wza and Wzc (**Fig. 1a**)”

Line 87: Please explain why the ‘original ribosome binding site’ was maintained in the expression constructs.

We rephrased the text accordingly.

Results section, p. 6: “We reconstructed the *wzabc* operon under a T7 promoter with the original ribosome binding site sequences to maintain a near-native protein expression ratio, and introduced a Strep-tag at the C-terminus of Wza.”

Magnesium occupancy of the ADP binding sites: is it possible that the Mg²⁺ ion is not resolved in some states due to heterogeneity? This could be tested by symmetry expansion and careful sorting of the protomers.

It is unlikely that Mg²⁺ ions remained bound following treatment with 20 mM EDTA, particularly since size-exclusion chromatography was conducted using a running buffer containing 10 mM EDTA. Therefore, we consider it improbable that the lack of observable Mg²⁺ density is due to structural heterogeneity among the protomers.

Line 141-142: It is proposed that the Wzc conformational transitions may generate a mechanical force that could translocate the CPS. First, this idea could be moved to the Discussion and second, it would require some supporting information. For example, are there any sites inside the tunnel that could interact with the CPS, and if so, are these sites affected by the structural reorientations? Can the authors compare the electrostatic properties of the channel interior in all observed states? Also, a careful comparison of conformation 0 and IV could reveal critical transitions associated with Wza binding.

Our proposed idea on the mechanical force has now been moved to the **Discussion** section on p. 15 as suggested. Since the CPS polymer often exhibits variable lengths, it is challenging to obtain a uniform substrate for detailed interaction studies. Given that Conf 0 is nearly identical to Conf I, and Conf V does not form a continuous channel, we focused our analysis on comparing the architecture and electrostatic properties of the periplasmic domain of Wzc across Confs I to IV. These analyses provide additional insights into the structural changes within the channel that may facilitate CPS translocation.

Line 197: ‘Ion leakage’ Where would the ions come from, and which membrane would they leak across? The outer membrane should be permeable to most small solutes and ions. Is there evidence that Wzc conducts ions across the inner membrane?

Wzc and Wza octamers are not always co-localized, suggesting that isolated Wzc octamers may also exist in vivo. However, previously reported structures of the *E. coli* K30 Wzc octamer (DOI: 10.1038/s41467-021-24652-1; DOI: 10.1038/s41467-025-58693-7) revealed a central channel approximately 30 Å in diameter. An opening of this size at the inner membrane would allow uncontrolled leakage of small ions, metabolites, and even folded peptides from the cytoplasm into the periplasm, which would compromise cell viability—contradicting the known tolerance of *E. coli* to Wzc overexpression. Thus, under physiological conditions, the central cavity of Wzc must be closed. In this study, we propose that our *E. coli* K12 Wzc model, which exhibits a sealed inner membrane cavity, more accurately represents the physiologically relevant conformation. To avoid potential confusion, we have revised the text accordingly.

Results section, p. 11: “Compared to the previously reported *E. coli* K30 Wzc_{K540M} structure with a wide open inner membrane cavity (3.5 nm in diameter)^{13,29}, the *E. coli* K12 Wzc_{K540M} exhibited a C4 symmetric conformation, with the inner membrane cavity tightly sealed by the forearms in Conf i (Fig. 4h and Supplementary Fig. 7g, h). This “closed” conformation of the Wzc octamer likely represents its native *in vivo* state, ensuring that the central cavity remains impermeable to ions and small molecules before association with Wza, thereby maintaining membrane integrity and electrochemical balance.”

Conclusion:

Discussion, line 316: The authors state that Wzc exists in a monomer-oligomer equilibrium when phosphorylated. However, the presented data do not address this point. If this statement is based on previously published data, this should be clarified. To address this point, could the Y-tail be removed or replaced, followed by fluorescent imaging.

We agree that the statement concerning the monomer–oligomer equilibrium of Wzc should be supported by appropriate references. In our study, live-cell imaging showed that Wzc exhibits both evenly distributed and clustered patterns on the membrane. In addition, biochemical analyses of the purified full-length Wzc revealed heterogeneous oligomeric states and variable phosphorylation levels, as determined by mass spectrometry. These findings suggest that Wzc exists in a phosphorylation-dependent monomer–oligomer equilibrium.

This interpretation is consistent with previous studies. The cytoplasmic domain of Wzc (DOI: 10.1074/jbc.M113.457804) has been reported to be monomeric when phosphorylated. Conversely, mutation of the Walker A motif lysine (K540M in Wzc), which mimics the non-phosphorylated state, promotes stable octamer formation (DOI: 10.1038/s41467-021-24652-1). Moreover, the phosphomimetic WzcK540M5YE variant fails to form a stable octamer (DOI: 10.1038/s41467-021-24652-1). Collectively, these published findings, together with our experimental data, support the conclusion that Wzc undergoes a phosphorylation-dependent transition between monomeric and oligomeric states.

This reference (DOI: 10.1038/s41467-021-24652-1) was cited in support of our conclusions.

Discussion, last paragraph, lines 313 - 328: Please rephrase to emphasize that this paragraph presents a hypothetical model.

The paragraph has been revised to clearly indicate that it presents a hypothetical model.

Discussion section, p. 17-18: “Our findings support a proposed model for CPS polymerization and export (Fig. 6), which, while hypothetical, provides a conceptual framework for understanding the underlying processes. According to this model, This model offers a

foundation for future experimental investigation into the mechanisms governing CPS biosynthesis and transport.”

Methods: Description of cryo-EM data processing, lines 541-543: This section is fairly short, which is ok if the data processing was straightforward. If not, technical difficulties related to particle picking, sorting, or focused refinements should be described.

In our case, data processing was relatively straightforward due to the large particle size, which facilitated efficient particle picking using CryoSPARC’s blob-picker. While focused refinements could enhance the resolution of local maps, we opted not to include these refinements in the model-building process. This decision was made to maintain consistency across the Wzc–Wza complexes in different conformations, ensuring direct comparability of the resulting structures.

Methods, glycan array, line 545 – 547: Please state the vendor and catalogue number of the arrays used.

We thank the reviewer for the suggestion. The information about the vendor and catalogue number of the glycan array was included in the revised manuscript.

Methods section, p. 28: “Glycan array screening was conducted using the Glycan Array 100 kit purchased from RayBioTech (Catalogue #: GA-Glycan-100-1; Norcross, GA, USA).”

REVIEWER COMMENTS

Reviewer #2 (Remarks to the Author):

The revised manuscript by Biao Yuan et al. has improved significantly. The authors have addressed most of our concerns, although some remain.

We thank reviewer #2 for the positive evaluation of our revised manuscript and for acknowledging the significant improvements made. We appreciate that most of the previous concerns have been addressed. We have now carefully considered the remaining comments and have provided detailed responses and additional revisions in the updated version of the manuscript to fully resolve these issues.

1. The co-localization analysis of Wza and Wzc is still not ideal. Including the D1 mutant was a good idea, but the concern regarding the overexpression of the components remains. To this end, the authors titrated IPTG to affect the expression levels of the binding partners. How do we know that the expression levels are indeed reduced at 10 versus 40 μ M IPTG? The authors state that 10 μ M corresponds to approx. endogenous expression levels, but how do we know?

We thank the reviewer for this valuable comment regarding the expression levels of Wza and Wzc in the co-localization assay. We acknowledge that our current experimental setup does not allow us to directly determine whether the expression at 10 μ M IPTG is close to the true endogenous expression level. This is why we avoided any claims about endogenous expression in our previous revised manuscript.

We have now included Supplementary Fig. 1a,b to show a clear reduction in the total fluorescence intensity per bacterial cell when comparing samples induced with 10 μ M versus 40 μ M IPTG, clearly suggesting a lower expression level at reduced IPTG concentration. Still we cannot exclude that the system still operates under overexpression conditions even at 10 μ M IPTG.

However, we have also conducted a control assay with the D1 mutant under identical experimental conditions, as suggested by Reviewer #2, and clearly observed that the co-localization between Wza and Wzc was disrupted. Additionally, a similar trend for the co-localization between Wza and Wzc has been observed under lower IPTG concentrations (see above). Therefore we conclude that these findings strongly support our claim that the observed interaction is specific and not an artifact due to overexpression. Consequently, whether the expression at 10 μ M IPTG is close to

endogenous or remains slightly elevated does not alter the interpretation or validity of our results.

We have incorporated the reduced expression level at 10 μM IPTG in the Result section.

Result section: p. 5: “Although expression was reduced at lower IPTG concentration (Supplementary Fig. 1a,b), a similar trend was observed under both conditions, demonstrating that the colocalization is not attributable to overexpression (**Fig. 1c**).”

Supplementary information: Supplementary Fig. 1, updated figure legend, p.2: “Single-cell quantification of protein expression after IPTG induction (**a,b**). Following induction with IPTG at the indicated concentrations for 16 hours at 19 $^{\circ}\text{C}$, we imaged the cells using confocal fluorescence microscopy (**a**); Scale bar: 5 μm . To quantify the

fluorescence signal per cell, the signal of Wzc-sfGFP was first used to create a binary mask in Fiji/ImageJ, which was then used to segment individual cells from each micrograph. The integrated intensity per individual cell was then quantified for each cell in both channels using Fiji/ImageJ. The results are shown in **b**.”

2. Mutational analysis of Wzc: We recommended to include a comparison of the expression levels of the individual Wzc mutants. To this end, the authors included a Western blot as a supplemental figure (1d). However, it is not stated what antibody was used and, more importantly, how the samples were normalized. Was this based on optical density? Ideally, one would include a blot for a housekeeping protein as a loading control.

We sincerely apologize for the oversight in the previous version, where the Western blot analysis was not described in the Methods section. In the revised manuscript, we have now included a detailed description of the experimental procedure.

Specifically, we used the Strep-tactin AP conjugate (now specified in the Methods section) and ensured that equal amounts of total cells were loaded for each sample, normalized according to their OD₆₀₀ values, to allow reliable comparison of protein expression levels among the Wzc mutants.

While we acknowledge that including a housekeeping protein as a loading control would be ideal, we are convinced that normalization by OD₆₀₀ for the same bacterial strain harboring different plasmids under identical experimental conditions ensures a consistent and appropriate comparison as well.

The corresponding information has been added to both the Methods section and the legend of Supplementary Fig. 1f.

Methods section, p. 31-32: **“Western-blot analysis**

Cell cultures from the colanic acid production assay were normalized according to their OD₆₀₀ values, ensuring equal cell numbers across samples. Equivalent cell aliquots were then mixed with SDS-PAGE loading buffer and incubated at 95 °C for 15 min to denature proteins. Subsequently, 10 µl of each sample was loaded onto an SDS-PAGE gel and electrophoresed at 200 V for 30 min.

Following electrophoresis, proteins were semi-dry transferred onto PVDF membranes at 25 V for 30 min using a Trans-Blot Turbo Transfer System (Bio-Rad). The membrane was then blocked for 30 min in a blocking buffer containing 3% bovine serum albumin (BSA) in 1 × TBST (20 mM Tris-HCl, pH 7.5; 150 mM NaCl; 0.1% [v/v] Tween-20). After

blocking, the membranes were washed three times with 1 × TBST and incubated at room temperature for 2 hours with Strep-Tactin AP Conjugate (IBA Lifesciences GmbH) diluted 1:4000 in TBST.

Following incubation, the membranes were washed again three times with 1 × TBST. For colorimetric detection, 5 ml of freshly prepared alkaline phosphatase (AP) buffer (100 mM Tris, pH 8.0; 150 mM NaCl; 1 mM MgCl₂) containing 33 μl NBT and 66 μl BCIP was added to each membrane and incubated until visible color development occurred. The reaction was terminated by rinsing the membranes with distilled water to prevent nonspecific background. Finally, the membrane was scanned for documentation.”

Supplementary information, Supplementary figure legend p.2: “(f) Western blot analysis of Wzc variant expression levels corresponding to the samples used in the CPS production assay. Cell cultures were normalized to equal cell densities based on OD₆₀₀ measurements during SDS-PAGE sample preparation. Strep-Tactin AP conjugate was used to detect the Wzc-strep variants.”

3. JR domain binding to colanic acid. Based on our suggestion, the authors tested whether the JR domain binds to colanic acid on the cell surface but failed to detect binding. We appreciate that this was attempted. However, this information is not included or discussed as part of the model. It seems that microarray data suggest binding to certain motifs, yet the observation that the JR domain does not bind to the assembled CPS may tell us something about what it really recognizes. Therefore, we suggest including this negative data in the overall description of carbohydrate binding to the JR domain.

As suggested, we have now included the method into the Methods section and discussed this negative data and its interpretation in the Discussion section of the revised manuscript, emphasizing that while the JR domain shows affinity for specific carbohydrate motifs in microarray analyses, it does not appear to bind strongly to the fully assembled colanic acid.

c

Supplementary information: Figure legend of Supplementary Fig. 9, p16.

(c) Fluorescence intensity of *E. coli* after co-sedimentation with sfGFP-Wzc_{JR}. WT: *E. coli* JM109 (DE3) cells. $\Delta wzabc$: *E. coli* JM109 (DE3) $\Delta wzabc$ cells, which lack colanic acid production. Measurements were performed in triplicate.

Methods section, p. 34:

“Co-sedimentation assay

To evaluate the ability of the sfGFP-Wzc_{JR} fusion protein to bind colanic acid-producing *E. coli*, a co-sedimentation assay was performed using *E. coli* JM109 (DE3) and the isogenic colanic acid-deficient strain JM109 (DE3) $\Delta wzabc$. 1 ml of each cell culture from CPS production assay was harvested by centrifugation (11,000 × g, 1 min). Cell pellets were resuspended in 1× PBS, and purified sfGFP-Wzc_{JR} was added to the cell suspensions at a final concentration of 0.1 mg/ml. Samples were incubated for 10 min at room temperature without shaking to allow binding. Following incubation, cells were collected by centrifugation (11,000 × g, 1 min) and washed twice with 1× PBS to remove unbound protein. The final pellets were resuspended in 1 ml of 1× PBS, and 100 μl of each suspension was transferred into a 96-well microplate. Fluorescence was measured using a TECAN plate reader with an excitation wavelength of 488 nm and emission at 535 nm. Optical density at 600 nm (OD₆₀₀) was measured from the same wells. Fluorescence values were normalized to OD₆₀₀ to account for variations in cell density.”

Discussion section, p.17: “The JR domain exhibited affinity toward specific carbohydrate motifs based on the glycan microarray data, but failed to interact strongly with the intact colanic acid polymer, as evidenced by the negative results from our cosedimentation assay using colanic acid-producing *E. coli* JM109 (DE3) with purified sfGFP-Wzc_{JR} (Supplementary Fig. 9c). This suggests that the structural complexity and higher-order organization of the polysaccharide might critically influence molecular recognition. These findings imply that the binding epitopes identified in vitro may not be readily accessible or appropriately presented within the native polymer matrix. Future studies will be required to quantitatively characterize the binding affinity of the JR domain toward both individual colanic acid repeat units and the fully assembled polymer.”

4. Prompted by our skepticism, the authors now state that the dimeric Wza complex associated with one Wzc is unlikely to be an *in vitro* artefact. Apparently, one supporting argument is that Wza tends to form 2D lattices. However, the arrangement of Wza in conformation V is inconsistent with a planar 2D lattice because the copies are not coplanar. Thus, we still caution overinterpreting the biological significance of conformation V.

We fully appreciate the suggested caution regarding the biological significance of conformation V. We agree that, although our cryo-EM data show the presence of a dimeric Wza complex associated with a single Wzc molecule, this arrangement should be interpreted with care.

We acknowledge that the non-coplanar orientation of Wza in conformation V is inconsistent with a perfectly planar 2D lattice and therefore does not directly support a stable lattice-like organization *in vivo*. In the revised manuscript, we have accordingly tempered our interpretation, noting that this conformation may represent a transient interaction rather than a stable physiological complex, or potentially an *in vitro* artifact arising during purification.

Nevertheless, we are convinced that the observed association between one Wzc and two Wza translocons in conformation V provides valuable insight into the assembly process, specifically, that four helical arms of Wzc are sufficient to recruit the Wza translocon.

We have revised the relevant section of the Discussion to reflect this more balanced interpretation and to avoid any overstatement of biological significance.

Discussion section, p.15-16: “An intriguing aspect of our study is the identification of a Wzc octamer associated with two Wza translocons (Conf V). This configuration may represent an *in vitro* complex formed during isolation. Alternatively it could be due to an EDTA-induced dissociation and subsequent artifactual re-association of the Wza-Wzc complex. Nevertheless, the observed arrangement suggests that four helical arms of the Wzc octamer are sufficient to recruit a Wza translocon. Given the known tendency of Wza to form higher-order clusters *in vivo* (Fig. 1d), Conf V may correspond to a transient intermediate that facilitates dynamic modulation between Wzc and Wza, thereby enhancing CPS secretion efficiency. However, further *in vivo* investigations and high-resolution imaging approaches such as cryo-ET and MINFLUX nanoscopy will be important to determine whether Conf V naturally occurs in bacterial systems.”